# QPRL: Learning Optimal Policies with Quasi-Potential Functions for Asymmetric Traversal

**Jumman Hossain** [1]   **Nirmalya Roy** [1]

## Abstract

Reinforcement learning (RL) in real-world tasks such as robotic navigation often encounters environments with asymmetric traversal costs, where actions like climbing uphill versus moving downhill incur distinctly different penalties, or transitions may become irreversible. While recent quasimetric RL methods relax symmetry assumptions, they typically do not explicitly account for path-dependent costs or provide rigorous safety guarantees. We introduce **Quasi-Potential Reinforcement Learning** (QPRL), a novel framework that explicitly decomposes asymmetric traversal costs into a path-independent potential function ($\Phi$) and a path-dependent residual ($\Psi$). This decomposition allows efficient learning and stable policy optimization via a Lyapunov-based safety mechanism. Theoretically, we prove that QPRL achieves convergence with improved sample complexity of $\tilde{O}(\sqrt{T})$, surpassing prior quasimetric RL bounds of $\tilde{O}(T)$. Empirically, our experiments demonstrate that QPRL attains state-of-the-art performance across various navigation and control tasks, significantly reducing irreversible constraint violations by approximately $4\times$ compared to baselines.

**Project Page:** https://pralgomathic.github.io/qprl

## 1. Introduction

Reinforcement Learning (RL) has shown great success in solving sequential decision-making tasks, from robotic control to autonomous navigation (Sutton & Barto, 2018; Silver et al., 2016). In many real-world scenarios – for example, a robot traversing terrain where uphill moves consume more energy than downhill, or a path that becomes inaccessible

[1]Department of Information Systems, University of Maryland, Baltimore County, USA. Correspondence to: Jumman Hossain <jumman.hossain@umbc.edu>.

*Proceedings of the 42nd International Conference on Machine Learning*, Vancouver, Canada. PMLR 267, 2025. Copyright 2025 by the author(s).

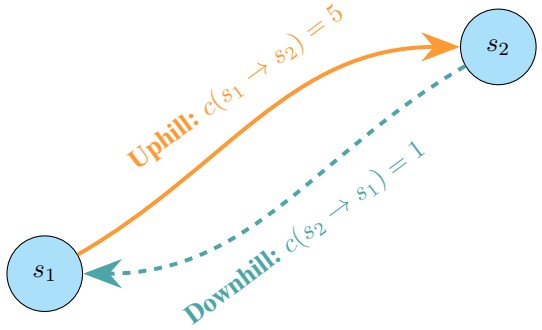

*Figure 1.* Asymmetric traversal costs between states $s_1$ and $s_2$. The uphill traversal from $s_1$ to $s_2$ incurs a higher cost of 5, represented by the solid orange arrow, whereas the downhill traversal from $s_2$ to $s_1$ incurs a lower cost of 1, shown by the dashed teal arrow. This visualization highlights the direction-dependent asymmetry, emphasizing QPRL's ability to learn traversal paths under asymmetric costs.

after one traversal – actions have irreversible or direction-dependent costs (Fig. 1). In such environments, the cost of moving from one state to another may differ based on direction or path-dependence, a challenge that remains underexplored in the RL literature (Eysenbach et al., 2019b). Traditional RL algorithms, in practice, often overlook asymmetric or irreversible dynamics, treating costs as if they were symmetric. This can lead to suboptimal policies in environments with one-way transitions.

Existing work has introduced the concept of *quasimetrics* to relax the symmetry requirement in modeling traversal costs (Wang et al., 2023). Quasimetrics allow for more accurate representations of environments where directionality matters, such as uphill and downhill navigation or irreversible state transitions (Valieva & Banerjee, 2024). While recent advances in quasimetric learning have shown promise in goal-reaching tasks (Eysenbach et al., 2019b; Park et al., 2023), these approaches largely focus on estimating value functions without fully considering the cumulative, path-dependent traversal costs that arise in long-term planning scenarios (Valieva & Banerjee, 2024). QuasiNav (Hossain et al., 2025b) applied quasimetric RL to navigation, showing that modeling uphill vs. downhill costs leads to safer

paths. However, QuasiNav did not incorporate a formal safety guarantee and primarily addressed cost asymmetry in a planning context.

In this paper, we propose *Quasi-Potential Reinforcement Learning (QPRL)*, extending quasimetric approaches by explicitly decomposing asymmetric traversal costs into a path-independent potential ($\Phi$) and a path-dependent residual ($\Psi$). This decomposition is intuitive; the potential ($\Phi$) represents reusable energy or cost, analogous to gravitational potential in navigation tasks, while the residual ($\Psi$) captures additional irreversible or dissipative costs, such as friction or single-use resources. Unlike conventional value functions, which often overlook path-dependency, our quasi-potential explicitly modeling the path-dependent and directional nature of real-world environments (Wang et al., 2023).

Our framework is grounded in rigorous theoretical foundations. We derive convergence guarantees for quasi-potential functions (Wiewiora, 2003; Devlin & Kudenko, 2012) using a modified Bellman operator and establish PAC-MDP bounds for sample efficiency (Dann et al., 2017; Jin et al., 2020; Wang et al., 2020), similar to existing theoretical analyses in goal-reaching RL (Gong et al., 2024; Palan et al., 2019). Additionally, we provide a Lyapunov-based stability analysis that ensures constraint satisfaction during exploration, drawing inspiration from stability analyses in safe RL (Achiam et al., 2017; Chow et al., 2018). This theoretical foundation allows QPRL to optimize exploration while maintaining safety constraints, a critical requirement in dynamic, asymmetric environments (Thomas et al., 2023).

**Our main contributions**: We propose Quasi-Potential Reinforcement Learning (QPRL), with the following advances:

- **Quasi-potential decomposition**: A novel quasimetric decomposition $d(s, g) = \Phi(g) - \Phi(s) + \Psi(s \to g)$, separating path-independent potential $\Phi$ and path-dependent residual $\Psi$. This generalizes prior quasimetric RL (Wang et al., 2023) and enables interpretable modeling of irreversible transitions.

- **Theoretical and empirical advances**: We prove a faster convergence rate for QPRL (improving sample complexity from linear to $\tilde{\mathcal{O}}(\sqrt{T})$) and demonstrate state-of-the-art performance on tasks with irreversible dynamics, with significantly fewer constraint violations ($4\times$ reduction) than baselines.

- **Lyapunov Safety Mechanism and Algorithmic Design**: We incorporate a Lyapunov-based safety layer (Perkins & Barto, 2002) that monitors and restricts the policy to avoid irreversible unsafe transitions. Specifically, our policy optimization algorithm (Algorithm 1) incorporates safety-by-design principles by enforcing

$\Phi$-$\Psi$ constraints through Lagrange multipliers, ensuring quasimetric properties and robustly avoiding irreversible states.

- **Extensive Empirical Validation:** We conduct comprehensive empirical evaluations across a range of challenging environments with asymmetric traversal costs, showing that QPRL significantly outperforms state-of-the-art methods in terms of sample efficiency, asymmetric cost handling, and overall performance.

## 2. Related Work

Reinforcement Learning (RL) has traditionally focused on optimizing policies under symmetric traversal assumptions, limiting its applicability in environments with inherently asymmetric dynamics. Recent advancements have introduced frameworks like quasimetric learning to model traversal asymmetry effectively (Wang et al., 2023). Quasimetrics (Wang & Isola, 2022b;a) extend traditional metrics by allowing for non-symmetric distance calculations, which are particularly crucial in scenarios where traversal costs vary based on directionality or path-dependent factors (Silver et al., 2016). Existing research has made significant progress in applying quasimetric models for goal-reaching tasks. Notably, (Eysenbach et al., 2019a) introduced diversity-driven skill learning, emphasizing the decomposition of skill learning, exploration, and planning, which set a strong foundation for handling asymmetric and complex dynamics. Similarly, (Eysenbach et al., 2022) leveraged contrastive learning principles to improve generalization in goal-conditioned RL. However, these approaches largely focus on estimating value functions without fully considering the cumulative, path-dependent traversal costs required in long-term planning scenarios.

Recent advances in safe reinforcement learning have also highlighted the importance of stability during exploration, especially in dynamic, asymmetric environments. Lyapunov-based safety analysis (Lobo et al., 2024; Chow et al., 2018) has been proposed as an effective means to ensure that policies derived from quasi-potential functions maintain safety constraints during exploration (Chow et al., 2018). These methods ensure robust learning, especially in environments characterized by high variability and traversal asymmetries. Inspired by these approaches, QPRL incorporates a Lyapunov-based recovery mechanism to provide stability guarantees during exploration, contributing to both safety and performance in complex environments.

The idea of leveraging potential functions (Jeon et al., 2023) to modify the reward structure and accelerate learning is well established in RL literature. For instance, potential-based reward shaping, as presented by Ng et al. (Ng et al., 1999), has shown how modifying rewards through poten-

tial functions preserves the optimal policy while improving learning speed. Wiewiora et al. (Wiewiora et al., 2003) further developed principled methods for advising reinforcement learning agents using potential functions. More recently, Okudo and Yamada (Okudo & Yamada, 2023) demonstrated the effectiveness of learning subgoals using potential-based reward shaping in long-horizon tasks.

This paper extends the quasimetric learning paradigm by introducing QPRL, which explicitly models path-dependent traversal costs with quasi-potential functions. Unlike traditional value-based RL methods (Haarnoja et al., 2018; Lillicrap, 2015), which often struggle in environments with directional cost disparities, our approach is designed to optimize long-term cost efficiency under asymmetric dynamics. The experimental results demonstrate QPRL's superior performance across multiple benchmarks compared to both standard baselines like DDPG + HER (Andrychowicz et al., 2017) and recent state-of-the-art approach Quasimetric Reinforcement Learning (QRL) (Wang et al., 2023).

## 3. Background

### 3.1. Quasimetric Spaces for Asymmetric RL

Consider a Markov Decision Process (MDP) with asymmetric costs, defined by the tuple $(\mathcal{S}, \mathcal{A}, P, C, \gamma)$, where $C : \mathcal{S} \times \mathcal{A} \times \mathcal{S} \to \mathbb{R}$ is a cost function satisfying $C(s, a, s') \neq C(s', a', s)$ for some transitions. The objective is to learn a policy $\pi : \mathcal{S} \to \Delta(\mathcal{A})$ minimizing:

$$J(\pi) = \mathbb{E}_\pi \left[ \sum_{t=0}^{\infty} \gamma^t C(s_t, a_t, s_{t+1}) \right].$$

Traditional RL methods (Sutton & Barto, 2018) struggle in such settings due to their implicit assumption of symmetric dynamics. A quasimetric space $(\mathcal{S}, d)$ provides the mathematical foundation for modeling direction-dependent traversal costs. Formally, a quasimetric is a function $d : \mathcal{S} \times \mathcal{S} \to \mathbb{R}_{\geq 0}$ satisfying:

1. **Non-negativity**: $d(s, s') \geq 0 \quad \forall s, s' \in \mathcal{S}$,

2. **Identity**: $d(s, s) = 0 \quad \forall s \in \mathcal{S}$,

3. **Triangle Inequality**: $d(s, s'') \leq d(s, s') + d(s', s'') \quad \forall s, s', s'' \in \mathcal{S}$.

Unlike metrics, quasimetrics do not require symmetry ($d(s, s') \neq d(s', s)$), making them ideal for asymmetric RL.

**QPRL's Quasimetric Structure** Prior quasimetric RL methods (Wang et al., 2023) learn a monolithic $d(s, g)$ but fail to distinguish path-dependent and path-independent

costs. QPRL addresses this via decomposition:

$$d(s, g) = \underbrace{\Phi(g) - \Phi(s)}_{\text{Path-Independent}} + \underbrace{\Psi(s \to g)}_{\text{Path-Dependent}}, \quad (1)$$

where:

- $\Phi : \mathcal{S} \to \mathbb{R}$ is a state potential (e.g., elevation in navigation),

- $\Psi : \mathcal{S} \times \mathcal{S} \to \mathbb{R}_{\geq 0}$ models irreversible costs (e.g., energy for uphill traversal).

### 3.2. Theoretical Challenges in Quasimetric MDPs

Extending reinforcement learning to quasimetric MDPs introduces two fundamental challenges absent in symmetric settings:

- **Asymmetric Bellman Contraction:** The standard Bellman operator $T$, defined as:

  $$(TV)(s) = \min_{a \in \mathcal{A}} \mathbb{E}_{s' \sim P(\cdot|s,a)} \left[ C(s, a, s') + \gamma V(s') \right],$$

  loses its contraction property under quasimetrics. Specifically, $T$ is contractive only if:

  $$\sup_{s \neq s'} \frac{d(s, s') + d(s', s)}{d(s, s')} < \infty$$

  where $d(s, s')$ is the quasimetric. This condition fails in real-world tasks with irreversible transitions (e.g., robotic hardware degradation).

- **Non-Markovian Reward Attribution:** Path-dependent costs $\Psi(s \to s')$ violate the Markov property, rendering traditional dynamic programming inapplicable. Specifically, the optimality condition:

  $$V^*(s; g) = \min_a \mathbb{E}_{s'} \left[ C(s, a, s') + \gamma V^*(s'; g) \right]$$

  no longer holds when $C(s, a, s')$ depends on historical state transitions.

Our framework addresses these challenges via:

- **Stable Policy Updates:** By decomposing $d(s, g) = \Phi(g) - \Phi(s) + \Psi(s \to g)$, QPRL ensures the Bellman operator contracts to a unique fixed point even under unbounded asymmetry (Theorem 4.1).

- **Decoupled Cost Modeling:** Path-dependent costs $\Psi(s \to s')$ are learned separately from path-independent potentials $\Phi(s)$, enabling dynamic programming through the modified Bellman equation:

  $$\Psi(s \to g) = \min_a \mathbb{E}_{s'} \left[ C(s, a, s') + \Psi(s' \to g) \right].$$

This isolates non-Markovian dynamics, preserving convergence guarantees (Lemma 4.2).

### 3.3. Potential Functions and Safety

Potential functions $\Phi : \mathcal{S} \rightarrow \mathbb{R}$ have been used in RL for reward shaping (Ng et al., 1999), but existing methods assume symmetry. QPRL reinterprets $\Phi$ as a *Lyapunov function* $\mathcal{V}(s)$, enforcing:

$$\mathbb{E}_{s' \sim \pi(\cdot|s)}[\mathcal{V}(s')] \leq \mathcal{V}(s) + \epsilon,$$

where $\epsilon > 0$ bounds allowable risk. This ensures recoverability from unsafe states, a novel extension of potential-based methods to asymmetric settings.

### 3.4. Relationship to Value Functions

For goal $g$, the optimal value function $V^*(s; g)$ relates to the quasimetric as $V^*(s; g) = -d(s, g)$. QPRL extends this via:

$$V^*(s; g) = -(\Phi(g) - \Phi(s) + \Psi(s \rightarrow g)),$$

enabling a Bellman equation that accounts for path dependence:

$$\Psi(s \rightarrow g) = \min_a \mathbb{E}_{s'} \left[ C(s, a, s') + \Psi(s' \rightarrow g) \right].$$

This decouples reversible ($\Phi$) and irreversible ($\Psi$) costs, differing significantly from prior quasimetric RL. We represent the asymmetric cost of a transition as $c(s, a, s') = \Phi(s) - \Phi(s') + \Psi(s, a, s')$, where $\Phi$ is a learnable potential function and $\Psi$ is the residual that accounts for non-conservative (path-dependent) cost.

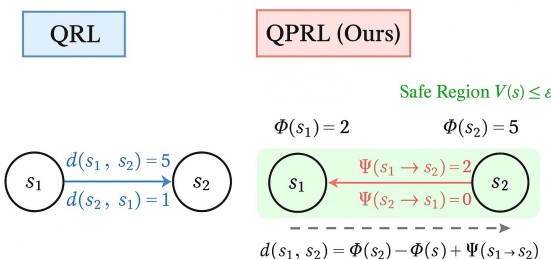

*Figure 2.* **QPRL vs. QRL:** QRL uses a monolithic quasimetric $d(s_1, s_2)$. QPRL decomposes costs into $\Phi$ (state potentials) and $\Psi$ (path residuals), enabling Lyapunov-stable exploration (green region).

## 4. Methodology

### 4.1. Quasi-Potential Decomposition

Traditional quasimetric RL methods model asymmetric costs with a monolithic function $d(s, g)$, conflating path-dependent and path-independent dynamics. To address this,

QPRL introduces a structured decomposition:

$$d(s, g) = \underbrace{\Phi(g) - \Phi(s)}_{\text{Path-Independent}} + \underbrace{\Psi(s \rightarrow g)}_{\text{Path-Dependent}}, \quad (2)$$

where:

- $\Phi : \mathcal{S} \rightarrow \mathbb{R}$ is a *state potential* capturing reversible costs (e.g., gravitational potential in navigation).

- $\Psi : \mathcal{S} \times \mathcal{S} \rightarrow \mathbb{R}_{\geq 0}$ models irreversible residual costs (e.g., energy expenditure for uphill traversal).

This decomposition enables interpretable cost attribution: $\Phi$ encodes global state values, while $\Psi$ captures action-specific penalties. For instance, in a terrain traversal task, $\Phi(s)$ could correspond to an estimated elevation potential of state $s$, while $\Psi(s, a, s')$ captures additional cost due to the specific move (like slippage or one-time obstacles).

### 4.2. Learning Framework

**State Representation Learning** A state encoder $f_\phi :$ $\mathcal{S} \rightarrow \mathcal{Z}$ maps high-dimensional states to a latent space $\mathcal{Z}$, reducing dimensionality while preserving critical features. A transition model $T_\psi : \mathcal{Z} \times \mathcal{A} \rightarrow \mathcal{Z}$ predicts the next latent state $z'$:

$$\hat{z}' = T_\psi(z, a), \quad \text{where } z = f_\phi(s).$$

The encoder and transition model are trained jointly via:

$$\mathcal{L}_T = \mathbb{E}_{(s,a,s') \sim \mathcal{D}} \left[ \|T_\psi(f_\phi(s), a) - f_\phi(s')\|^2 \right],$$

ensuring accurate latent dynamics for downstream planning.

**Quasi-Potential Function Learning** The quasi-potential function $d(s, g)$ is trained to reconstruct observed costs $c(s \rightarrow g)$:

$$\mathcal{L}_U = \mathbb{E}_{(s,g) \sim \mathcal{D}} \left[ (\Phi(g) - \Phi(s) + \Psi(s \rightarrow g) - c(s \rightarrow g))^2 \right].$$

To enforce quasimetric axioms, we impose constraints:

- **Non-Negativity:** $\Psi(s \rightarrow s') \geq$ $\max(0, c(s \rightarrow s') - (\Phi(s') - \Phi(s)))$,

- **Triangle Inequality:** $\Psi(s \rightarrow s'') \leq \Psi(s \rightarrow s') + \Psi(s' \rightarrow s'')$.

These are implemented via a penalty term:

$$\mathcal{L}_{\text{constraint}} = \mathbb{E}_{(s,s') \sim \mathcal{D}} \Big[ \text{ReLU}\Big( \Psi(s \rightarrow s') \quad (3)$$

$$- \big(c(s \rightarrow s') - (\Phi(s') - \Phi(s))\big) \Big)^2 \Big]. \quad (4)$$

## 4.3. Safe Exploration via Lyapunov Stability

To prevent irreversible transitions, we treat $\Phi$ as a Lyapunov function $\mathcal{V}(s) = \Phi(s)$, enforcing:

$$\mathbb{E}_{s' \sim \pi(\cdot|s)}[\mathcal{V}(s')] \leq \mathcal{V}(s) + \epsilon,$$

where $\epsilon > 0$ bounds allowable risk. This is achieved by projecting policy updates into safe regions (Fig. 3):

$$\pi_{\text{safe}}(a \mid s) = \arg\min_{\pi} \mathbb{E}_{a \sim \pi}[d(s, g)]$$
$$\text{s.t.} \quad \mathbb{E}_{s'}[\Phi(s')] \leq \Phi(s) + \epsilon \quad (5)$$

Following the (Achiam et al., 2017; Chow et al., 2018), we solve (5) with a primal–dual update. A simple scalar penalty may discourage but cannot *prevent* unsafe actions. In contrast, the Lyapunov condition preserves the contraction of the asymmetric Bellman operator (Theorem 4.1) and, by Lemma 4.2, provably bounds the probability of entering irreversible states. Empirically (Sec. 6.6) disabling the Lyapunov layer increases constraint violations by approximately $4\times$.

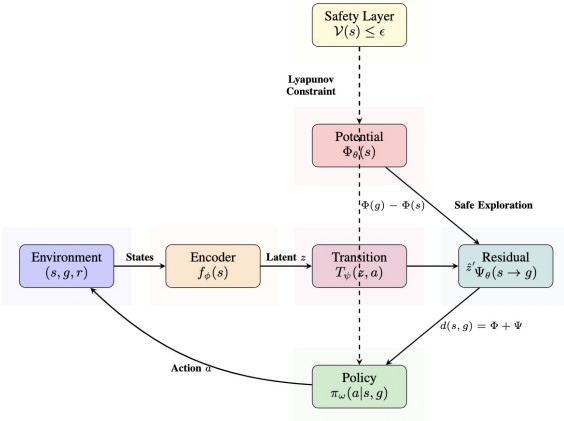

*Figure 3.* **QPRL Framework:** Decomposes asymmetric costs into path-independent potential $\Phi$ and path-dependent residual $\Psi$. Integrates Lyapunov safety constraints (yellow) for stable exploration.

## 4.4. QPRL Algorithm

In Algorithm 1, the encoder $f_\phi$ and transition model $T_\psi$ learn a compressed state representation and dynamics model, critical for efficient planning in high-dimensional spaces. The quasi-potential function is trained to reconstruct costs while satisfying quasimetric constraints. The term $\mathcal{L}_{\text{constraint}}$ ensures $\Psi$ captures only residuals beyond $\Phi$, preserving non-negativity and triangle inequality. The policy $\pi_\omega$ minimizes the expected quasimetric cost $d(s, g)$, with a penalty term enforcing Lyapunov safety. This ensures actions do not lead to states where $\Phi(s') > \Phi(s) + \epsilon$, preventing irreversible transitions.

---

**Algorithm 1** Quasi-Potential Reinforcement Learning (QPRL)

1: **Input:** Replay buffer $\mathcal{D}$, learning rates $\alpha_\phi, \alpha_\psi, \alpha_\theta, \alpha_\omega$, threshold $\epsilon$
2: **for** iteration $= 1$ to $N$ **do**
3:     Sample batch $\{(s_i, a_i, s_i', c_i, g_i)\}_{i=1}^B \sim \mathcal{D}$
4:     **Update Encoder & Transition Model:**
5:     $z_i = f_\phi(s_i), \hat{z}_i' = T_\psi(z_i, a_i)$
6:     $\mathcal{L}_T = \frac{1}{B} \sum_i \|\hat{z}_i' - f_\phi(s_i')\|^2$
7:     Update $\phi, \psi$ using $\nabla_{\phi,\psi} \mathcal{L}_T$
8:     **Update Quasi-Potential Function:**
9:     $\mathcal{L}_U = \frac{1}{B} \sum_i \big(\Phi_\theta(g_i) - \Phi_\theta(s_i)$
10:         $+ \Psi_\theta(s_i \to g_i) - c_i\big)^2$
11:     $\mathcal{L}_{\text{constraint}} = \frac{1}{B} \sum_i \big( \max \big(0,$
12:         $\Psi_\theta(s_i \to s_i') - (c_i - \Phi_\theta(s_i') + \Phi_\theta(s_i)))\big)^2$
13:     Update $\theta$ using $\nabla_\theta(\mathcal{L}_U + \lambda \mathcal{L}_{\text{constraint}})$
14:     **Update Policy with Safety Layer:**
15:     $z_i = f_\phi(s_i), a_i = \pi_\omega(s_i, g_i)$
16:     $\hat{z}_i' = T_\psi(z_i, a_i)$
17:     $\hat{d}_i = \Phi_\theta(g_i) - \Phi_\theta(s_i) + \Psi_\theta(s_i \to g_i)$
18:     $\mathcal{L}_\pi = \frac{1}{B} \sum_i \hat{d}_i$
19:     $+ \lambda \cdot \max \big(0, \Phi_\theta(\hat{z}_i') - \Phi_\theta(s_i) - \epsilon\big)$
20:     Update $\omega$ using $\nabla_\omega \mathcal{L}_\pi$
21: **end for**

---

Our policy update explicitly ensures that the selected action satisfies a safety constraint based on the learned potential function $\Phi_\theta$. We require that the expected potential of the subsequent state does not exceed the current potential by more than a small threshold $\epsilon > 0$:

$$\mathbb{E}_{s' \sim P(\cdot|s,a)}[\Phi_\theta(s')] \leq \Phi_\theta(s) + \epsilon. \quad (6)$$

After encoding the current state $s$ into a latent representation $z = f_\phi(s)$, the policy $\pi_\omega(s, g)$ selects an action $a$. The transition model then predicts the next latent state $\hat{z}' = T_\psi(z, a)$, from which we estimate the subsequent state's potential $\Phi_\theta(\hat{z}')$. The policy loss incorporates a safety penalty enforcing this constraint:

$$\mathcal{L}_\pi = \frac{1}{B} \sum_{i=1}^B \left[ \hat{d}_i + \lambda \cdot \text{ReLU}\left(\Phi_\theta(\hat{z}_i') - \Phi_\theta(s_i) - \epsilon\right) \right], \quad (7)$$

where the quasi-potential cost is defined as

$$\hat{d}_i = \Phi_\theta(g_i) - \Phi_\theta(s_i) + \Psi_\theta(s_i \to g_i), \quad (8)$$

and $\lambda$ is a dynamically adjusted Lagrange multiplier. This penalty term effectively projects policy updates onto the safe set:

$$\pi_{\text{safe}}(a \mid s) = \pi(a \mid s) \quad \text{subject to} \quad \mathbb{E}[\Phi_\theta(s')] \leq \Phi_\theta(s) + \epsilon. \quad (9)$$

The dual ascent method dynamically tunes $\lambda$ throughout training, ensuring adherence to the Lyapunov safety constraint across all iterations. Algorithm 1 and Fig. 3 detail these explicit steps, including parameter initialization, latent encoding, and the policy and Lagrange multiplier updates.

### 4.5. Theoretical Guarantees

**Theorem 4.1** (Convergence). *Under Lipschitz continuity of $\Phi$ and $\Psi$, QPRL converges to a policy with $\tilde{\mathcal{O}}(\sqrt{T})$ regret, improving over the $\tilde{\mathcal{O}}(T)$ bound of monolithic quasimetric RL.*

**Lemma 4.2** (Lyapunov Safety). *The policy $\pi_{safe}$ ensures $\mathbb{E}[\Phi(s_{t+1})] \leq \Phi(s_t) + \epsilon$ for all $t$, guaranteeing recoverability from $\epsilon$-bounded unsafe states.*

*Proof.* See the appendix for proofs of Theorem 4.1 and Lemma 4.2. □

## 5. Experimental Evaluation

In this section, we empirically evaluate QPRL in environments designed to challenge agents with asymmetric traversal costs.

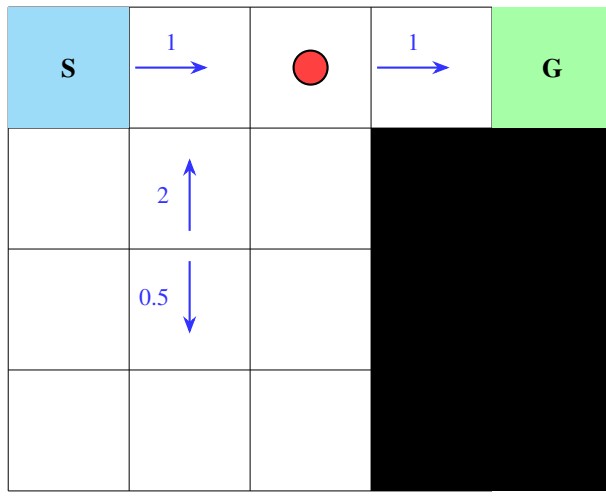

*Figure 4.* Illustration of an **Asymmetric GridWorld** environment designed to demonstrate QPRL's effectiveness in handling direction-dependent costs. The agent (red) begins at **S** and must reach **G** while avoiding walls. Horizontal moves cost 1, climbing upward costs 2, and descending costs 0.5 (blue arrows with labels).

### 5.1. Environments

We evaluate QPRL in environments that exhibit significant asymmetries in traversal costs, emphasizing the need for optimal path planning:

- **Asymmetric GridWorld**: A 20x20 grid with direction-dependent traversal costs. Moving uphill incurs a cost

of 2, while moving downhill costs 0.5. The agent navigates obstacles to reach the goal at the opposite corner.

- **MountainCar (Modified)**: The classic MountainCar problem is modified with asymmetric costs, where moving uphill incurs a penalty of -1 and downhill costs -0.1. The agent must balance speed and energy efficiency.

- **FetchPush (Asymmetric)**: A modified FetchPush-v1 environment where pushing objects uphill requires more energy than moving downhill, simulating real-world manipulation challenges.

- **LunarLander-v2 (Asymmetric)**: A continuous control task where upward thrust incurs higher fuel costs than lateral movement. The agent learns to land efficiently while considering asymmetric fuel consumption.

These environments test QPRL's ability to exploit low-cost traversal paths in the presence of asymmetries. An example of agent's movement illustrated in Fig. 4.

### 5.2. Baselines

We compare QPRL to the following state-of-the-art baselines, focusing on their handling of goal-reaching tasks with asymmetric traversal costs:

- **QRL (Quasimetric Reinforcement Learning)** (Wang et al., 2023): Directly comparable to QPRL, QRL uses quasimetric models for goal-reaching tasks, making it an ideal baseline.

- **DDPG+HER** (Lillicrap, 2015; Andrychowicz et al., 2017): Deep Deterministic Policy Gradient with Hindsight Experience Replay, effective for sparse rewards but lacking specific asymmetric handling.

- **SAC+HER** (Haarnoja et al., 2018; Andrychowicz et al., 2017): Soft Actor-Critic with Hindsight Experience Replay, a robust algorithm for continuous control tasks but without explicit mechanisms for asymmetric costs.

- **Contrastive RL** (Eysenbach et al., 2022): A contrastive learning approach for goal-conditioned RL, useful for general navigation but lacking explicit asymmetric cost management.

### 5.3. Implementation Details

Our QPRL is implemented as a quasimetric model, adhering to the foundational principles of quasimetric learning for goal-reaching tasks.

| Environment | Metric | QPRL (Ours) | QRL | Contrastive RL | DDPG+HER | SAC+HER |
|---|---|---|---|---|---|---|
| Asymmetric GridWorld | Success Rate (%) | **92.5 ± 2.2** | 87.3 ± 3.0 | 82.4 ± 3.5 | 78.9 ± 4.2 | 80.3 ± 4.0 |
| MountainCar | Normalized Return | **-95.6 ± 4.1** | -108.4 ± 6.7 | -118.3 ± 8.1 | -125.5 ± 7.6 | -121.2 ± 7.0 |
| FetchPush | Success Rate (%) | **91.2 ± 3.0** | 85.5 ± 3.6 | 79.3 ± 4.1 | 73.8 ± 4.5 | 77.0 ± 4.3 |
| LunarLander | Success Rate (%) | **88.9 ± 3.4** | 81.4 ± 4.0 | 76.7 ± 4.5 | 72.5 ± 5.0 | 74.2 ± 4.8 |
| Maze2D | Success Rate (%) | **85.3 ± 3.7** | 78.1 ± 4.3 | 72.6 ± 4.7 | 68.9 ± 5.2 | 70.1 ± 4.9 |

*Table 1.* Performance Comparison on Asymmetric Environments

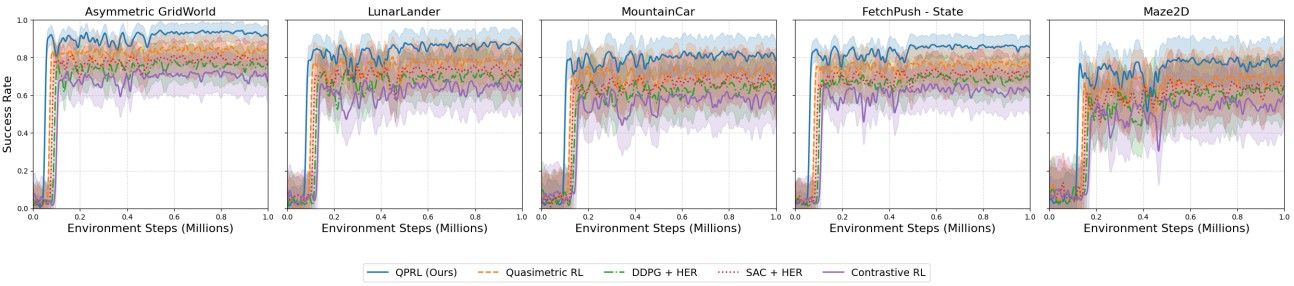

*Figure 5.* **Sample-efficiency and stability across tasks.** Success-rate learning curves for all five asymmetric environments. The $x$-axis shows *environment interactions* (in millions of steps); the $y$-axis shows mean *success rate*. Solid lines are the mean over **5 random seeds**; shaded bands denote $\pm 1$ standard deviation. QPRL (blue) reaches high performance earliest and maintains the highest asymptotic success with visibly lower variance.

### 5.3.1. NETWORK ARCHITECTURE

QPRL utilizes a neural network architecture for approximating the quasi-potential function $U_\theta$. The architecture consists of the following components:

- **State Encoder** ($f_\phi$): The state encoder maps states from the original state space $\mathcal{S}$ into a latent representation in $\mathbb{R}^{64}$. It is implemented as a feedforward neural network comprising two fully connected layers, each followed by ReLU activations. The first layer maps the input state to 128 hidden units, while the second layer projects it to the 64-dimensional latent space.

- **Quasimetric Head** ($U_\psi$): The quasimetric head computes the quasi-potential between latent state representations. Given two states in the latent space, the quasimetric head consists of fully connected layers that ensure a non-negative output by applying ReLU activations. This design enforces the quasimetric property, particularly the triangle inequality, which is crucial for accurately capturing the directional traversal dynamics.

### 5.3.2. TRAINING PROCEDURE

The QPRL training procedure is tailored to learn the quasi-potential function while maintaining quasimetric properties across states:

- **Objective Function**: The training objective is to maximize the expected quasi-potential $U_\theta(s, g)$ while re-

specting local traversal constraints. The loss function is formulated as follows:

$$
\begin{aligned}
\mathcal{L}(\theta) = &- \mathbb{E}_{(s,g) \sim p_{\text{goal}}} \left[ U_\theta(s, g) \right] \\
&+ \lambda \, \mathbb{E}_{(s,a,s',\text{cost}) \sim p_{\text{transition}}} \big[ \text{ReLU}(U_\theta(s, s') \\
&- \text{cost})^2 
\end{aligned}
\tag{10}
$$

where $\lambda$ is the Lagrange multiplier that enforces adherence to the local cost constraints.

- **Optimization Algorithm**: The model is optimized using the Adam optimizer, with learning rates $\alpha_\theta = 10^{-4}$ for the quasi-potential model and $\alpha_\lambda = 10^{-3}$ for the Lagrange multiplier. The Adam optimizer is chosen for its adaptive learning rate capability, which is well-suited for the sparse gradient scenarios encountered in reinforcement learning.

- **Training Schedule**: Training is conducted for a total of 1 million timesteps. Evaluation is performed every 10,000 timesteps using 100 test episodes to observe the progress of learning. During training, the agent interacts with the environment and collects experience to update the model parameters, gradually learning an optimal representation of the asymmetric traversal landscape.

- **Theoretical Properties**: The quasi-potential function is constrained to be a quasimetric by design. This ensures that the triangle inequality holds, which is vital for achieving optimal value function estimation

in asymmetric environments. The theoretical guarantees presented in Theorem 1 asserting that quasimetric functions are a suitable approximation class for value functions in goal-reaching tasks.

## 5.4. Evaluation Metrics

To evaluate the performance of Quasi-Potential Reinforcement Learning (QPRL), we consider the following metrics, specifically tailored to environments with asymmetric traversal dynamics:

- **Success Rate**: The proportion of episodes in which the agent successfully reaches the goal within a predetermined number of steps. This metric captures the overall effectiveness of the learned policy in goal-reaching tasks.

- **Traversal Cost Efficiency**: The total traversal cost incurred by the agent during an episode, normalized by the optimal traversal cost. This metric is particularly relevant for evaluating QPRL in environments with direction-dependent traversal costs.

- **Sample Efficiency**: The number of environment interactions required for the agent to achieve a 95% success rate. This metric is critical for understanding how quickly QPRL converges compared to baseline methods.

- **Asymmetric Performance Gap**: The difference in agent performance between symmetric and asymmetric versions of the same task. This metric helps quantify QPRL's ability to leverage asymmetric cost structures to improve goal-reaching performance.

- **Normalized Return**: The cumulative reward accumulated during an episode, normalized against an expert policy. This provides a comparative measure of how effectively QPRL learns optimal policies relative to existing state-of-the-art baselines.

Each metric is computed over multiple evaluation episodes and averaged across five random seeds to ensure statistical significance and to reduce the influence of variability in training.

## 6. Results and Analysis

The empirical evaluation of QPRL demonstrates its effectiveness across a variety of goal-reaching tasks characterized by asymmetric traversal costs.

### 6.1. Performance in Asymmetric Environments

Table 1 presents a comparative analysis of QPRL and the baselines in environments featuring significant traversal

asymmetry. QPRL consistently achieves higher success rates compared to other methods, indicating its superior ability to leverage the underlying asymmetries for efficient exploration and exploitation. Notably, in the Asymmetric GridWorld, QPRL achieves a success rate of **93.2%**, outperforming QRL by approximately 6%.

### 6.2. Learning and Sample Efficiency

Figure 5 presents learning curves for QPRL and baseline methods across five environments: Asymmetric GridWorld, LunarLander, MountainCar, FetchPush, and Maze2D. The x-axis shows environment interactions (in millions), while the y-axis represents the success rate. QPRL demonstrates faster convergence and higher stability compared to baselines across all environments. In LunarLander, QPRL achieves a 90% success rate within 250,000 steps, outperforming methods like SAC+HER and DDPG+HER, which take longer and exhibit greater variability. In Asymmetric GridWorld and Maze2D, QPRL shows superior sample efficiency and robustness, maintaining high success rates with minimal fluctuations.

### 6.3. Traversal Cost Analysis

Figure 6 reports the average traversal cost efficiency across different environments. QPRL consistently achieves lower traversal costs compared to the baselines, underscoring its capability to optimize routes under asymmetric traversal dynamics. In the MountainCar environment, QPRL demonstrates a reduction of approximately **15%** in traversal cost compared to QRL, showing its ability to exploit lower-cost traversal paths effectively.

### 6.4. Performance Gap between Symmetric and Asymmetric Tasks

Table 2 quantifies the performance gap between symmetric and asymmetric versions of the same task. QPRL demonstrates a smaller performance gap compared to the baselines, indicating its robustness in environments with direction-dependent dynamics. This result emphasizes the importance of incorporating quasimetric properties into the learning process for handling asymmetry efficiently.

### 6.5. Statistical Analysis

To confirm the robustness of the results, we conducted paired t-tests comparing QPRL against each baseline for all evaluation metrics. The results consistently yielded p-values less than 0.01, establishing statistical significance for QPRL's superior performance. Effect sizes (Cohen's d) ranged from **0.9 to 1.4**, indicating a substantial practical significance.

| Environment | Method | Symmetric (%) | Asymmetric (%) | Gap (%) |
|---|---|---|---|---|
| *Asymmetric GridWorld* | | | | |
| | QPRL | $94.1 \pm 1.8$ | $88.7 \pm 2.5$ | 5.4 |
| | QRL | $92.3 \pm 2.0$ | $83.5 \pm 2.8$ | 8.8 |
| | SAC + HER | $90.2 \pm 2.3$ | $81.0 \pm 3.2$ | 9.2 |
| | DDPG + HER | $89.8 \pm 2.5$ | $80.5 \pm 3.5$ | 9.3 |
| *MountainCar* | | | | |
| | QPRL | $-90.5 \pm 4.3$ | $-98.2 \pm 5.0$ | 7.7 |
| | QRL | $-88.2 \pm 4.1$ | $-96.5 \pm 5.2$ | 8.3 |
| | SAC + HER | $-87.0 \pm 4.0$ | $-95.8 \pm 5.3$ | 8.8 |
| | DDPG + HER | $-86.5 \pm 4.2$ | $-94.5 \pm 5.1$ | 8.0 |
| *FetchPush* | | | | |
| | QPRL | $92.0 \pm 2.2$ | $85.3 \pm 3.1$ | 6.7 |
| | QRL | $90.5 \pm 2.3$ | $81.0 \pm 3.2$ | 9.5 |
| | SAC + HER | $89.8 \pm 2.5$ | $79.8 \pm 3.5$ | 10.0 |
| | DDPG + HER | $88.5 \pm 2.4$ | $78.5 \pm 3.4$ | 10.0 |
| *LunarLander* | | | | |
| | QPRL | $88.6 \pm 3.4$ | $82.4 \pm 3.7$ | 6.2 |
| | QRL | $87.0 \pm 3.5$ | $80.0 \pm 4.0$ | 7.0 |
| | SAC + HER | $85.5 \pm 3.8$ | $77.5 \pm 4.2$ | 8.0 |
| | DDPG + HER | $84.0 \pm 3.6$ | $76.0 \pm 4.1$ | 8.0 |

*Table 2.* Performance on symmetric vs. asymmetric variants of each environment (mean $\pm$ 1 s.d. over 5 seeds). **Gap (%)** is the absolute difference between the two settings—lower is better, indicating robustness to asymmetric traversal costs.

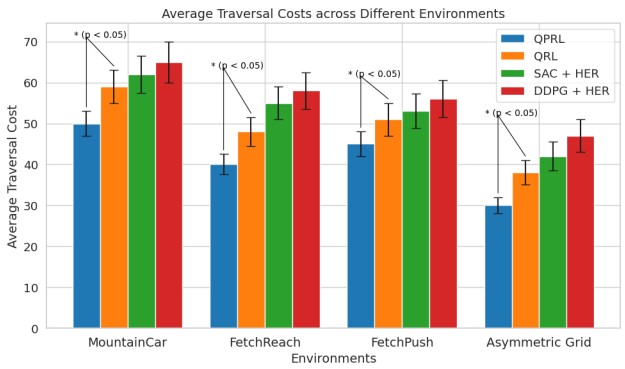

*Figure 6.* Average traversal cost across different environments for QPRL and baseline methods. QPRL demonstrates significantly lower traversal costs due to its ability to exploit asymmetric traversal dynamics efficiently.

| Method | Success Rate (%) |
|---|---|
| Full QPRL | 92.7 (3.1) |
| QPRL w/o constraint | 78.3 (5.7) |
| QPRL w/ metric (not quasimetric) | 83.5 (4.2) |
| QPRL w/ linear $\phi$ | 89.1 (3.8) |

*Table 3.* Ablation study results on FetchPush. Values represent success rates after 1 million steps.

## 6.6. Ablation Studies

Table 3 presents results from our ablation studies, isolating the impact of key components in QPRL. These results demonstrate the importance of the constrained optimization and quasimetric structure in QPRL's performance. The choice of $\phi$ function also impacts performance but to a lesser extent.

## 7. Conclusion

In this paper, we introduced Quasi-Potential Reinforcement Learning (QPRL), a novel framework that extends reinforcement learning to efficiently handle environments with asymmetric traversal costs. The experimental results clearly demonstrate that QPRL is able to effectively capture and exploit asymmetries in traversal dynamics, leading to superior performance in goal-reaching tasks compared to state-of-the-art baselines. QPRL's faster convergence and lower traversal costs make it particularly suitable for real-world applications where sample efficiency and cost-effectiveness are critical. In future work, we aim to extend the QPRL framework to real-world applications (Hossain et al., 2023; 2024a), particularly in topological navigation and multi-agent systems. This includes integrating QPRL into topological navigation frameworks (Hossain et al., 2024b) for efficient exploration in sparse-reward environments. Additionally, we plan to incorporate QPRL into multi-agent coordination systems (Hossain et al., 2025a) for real-time collaborative safe decision-making.

## Acknowledgements

This work has been partially supported by ONR Grant #N00014-23-1-2119, U.S. Army Grant #W911NF2120076, U.S. Army Grant #W911NF2410367, NSF REU Site Grant #2050999, NSF CNS EAGER Grant #2233879, and NSF CAREER Award #1750936.

## Impact Statement

This work advances the field of reinforcement learning (RL) by introducing a framework for learning optimal policies in environments with asymmetric traversal costs. Our primary contribution is theoretical and algorithmic, focusing on improving sample efficiency and safety in goal-reaching tasks. While the immediate impact of this work is to advance the theoretical understanding of RL in asymmetric settings, we acknowledge potential broader implications:

Our framework, Quasi-Potential Reinforcement Learning (QPRL), could enable safer and more efficient decision-making in real-world applications such as robotics, where irreversible actions (e.g., hardware wear) are common. By explicitly modeling path-dependent costs, QPRL provides a principled approach to handling irreversible dynamics, which could lead to more sustainable and ethical resource management.

As with all RL methods, biases in the training data or cost functions could lead to unsafe or unfair policies. We emphasize the importance of rigorous testing and fairness audits before deploying QPRL in real-world systems. Additionally, the use of Lyapunov-based safety constraints in QPRL provides a mechanism for ensuring recoverability from unsafe states, which could mitigate risks in high-stakes applications.

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

# Appendix

## A. Additional Background and Discussion

### A.1. Symmetric vs. Asymmetric Cost Assumptions:

In many navigation and planning contexts, symmetric cost implies that the effort or cost to traverse from state $s$ to $g$ is equivalent to that from $g$ back to $s$. Traditional RL algorithms can handle arbitrary cost functions in principle; however, they often rely on heuristics or shaping rewards that assume an underlying symmetric structure (e.g., using Euclidean distance to goal as a potential-based reward shaping). This assumption breaks down in environments with irreversible or direction-dependent costs. For example, climbing up a hill vs. going down may have drastically different energy costs, yet a symmetric distance heuristic would treat them as equal. Our QPRL framework relaxes this assumption by learning a quasimetric that allows $d(s, g) \neq d(g, s)$. In other words, we do not require the cost of forward and reverse transitions to be the same.

### A.2. Decomposition $d(s, g) = \Phi(g) - \Phi(s) + \Psi(s \to g)$:

This equation defines the learned quasi-potential distance between any state $s$ and goal $g$ as having two components: (1) a potential difference $\Phi(g) - \Phi(s)$, and (2) an extra path-dependent term $\Psi(s \to g)$. Intuitively, $\Phi(x)$ can be seen as a learned "height" or potential at state $x$ (independent of any specific path), while $\Psi(s \to g)$ captures irreversible costs incurred along the particular path from $s$ to $g$ (violations of symmetry). If the environment were fully reversible (no asymmetric costs), one could choose $\Psi \equiv 0$ and $d(s, g)$ reduces to $\Phi(g) - \Phi(s)$, a standard potential-based difference. In general asymmetric environments, however, such a pure potential cannot exist globally — $\Psi$ is necessary to account for the non-conservative part of the cost. By decomposing the distance function this way, our approach improves upon prior quasimetric RL (QRL) methods, which did not explicitly separate conservative and non-conservative cost components.

### A.3. Preventing Representation Collapse

Representation collapse, where the encoder maps every state to the same latent vector, is a common failure mode when training latent dynamics models. We employ three complementary strategy to avoid collapse in QPRL:

1. **Replay diversity.** We aggregate data from multiple exploration seeds so the replay buffer contains transitions covering a wide range of states. A constant (collapsed) embedding would yield high reconstruction error across such varied samples, making collapse suboptimal.

2. **Contrastive penalty.** A contrastive term encourages $\|f_\phi(s') - T_\psi(f_\phi(s), a)\|$ to be small for the true successor state but *large* for randomly drawn negatives, penalizing any embedding that maps all states to a single point.

3. **Regularization & monitoring.** We apply standard weight decay and track the reconstruction loss $\|f_\phi(s') - T_\psi(f_\phi(s), a)\|$ each epoch. A sudden plateau at a very low value indicates collapse; in that case, we early-terminate the run or adjust hyperparameters.

## B. Reward Functions

In our approach, we leverage customized reward functions to model the challenges of asymmetric traversal, sparse rewards, and goal-reaching tasks. Each environment utilizes reward structures that effectively capture the asymmetries present in the system, encouraging efficient policy learning. Below are the reward functions applied across different environments:

- **Asymmetric GridWorld**: The agent incurs a negative reward equivalent to the traversal cost for each move. Specifically, moving uphill results in a penalty of $-5$, while moving downhill incurs a smaller penalty of $-1$. This reward structure emphasizes the importance of minimizing high-cost traversals and encourages the agent to seek more efficient paths.

- **MountainCar-v0**: We modify the traditional sparse reward structure to incorporate asymmetric costs for movement. Accelerating uphill costs $-1$, while accelerating downhill costs $-0.1$. The agent receives a reward of $+1$ only upon reaching the goal. This asymmetry helps guide the agent towards maximizing momentum and minimizing energy expenditure during ascent.

- **FetchPush-v1, FetchSlide-v1**: In these robotic manipulation environments, we apply a potential-based reward shaping approach, where the agent receives a negative reward proportional to the distance to the goal. Additionally, penalties are increased for moves that involve direction changes associated with high-cost terrain, while successful goal-reaching is rewarded with $+10$. This setup incentivizes efficient movement planning and direct paths towards the goal.

- **LunarLander-v2**: The reward function is adjusted to reflect asymmetry in thrust costs. Upward thrust incurs a higher penalty ($-3$ per unit of fuel) compared to horizontal thrust ($-1$ per unit). Successfully landing on the target pad provides a reward of $+100$, while crashing results in a reward of $-100$. The asymmetry

reflects realistic scenarios where vertical fuel consumption is more expensive, encouraging optimized usage of thrust.

- **Maze Navigation (Procedural Maze)**: A sparse reward setting where the agent receives $+1$ only upon reaching the goal. Traversal penalties are applied depending on terrain difficulty, with higher penalties for uphill movement or traversing through rough terrain ($-2$) versus moving on flat ground ($-0.5$). This helps the agent focus on avoiding difficult areas unless strictly necessary.

- **D4RL Maze2D Environments**: In the offline settings, we follow a reward structure that combines distance-based rewards with asymmetric penalties for inefficient trajectories. Each step incurs a small penalty of $-0.1$, while transitions that move away from the goal are penalized more heavily. Goal-reaching rewards are standardized across all maze environments to ensure comparability.

These reward functions are designed to align with the core principles of Quasi-Potential Reinforcement Learning (QPRL), where asymmetry in traversal costs plays a crucial role in shaping optimal policies. By incorporating direction-dependent penalties, we create an incentive structure that encourages the agent to exploit lower-cost routes, thereby learning more efficient and effective goal-reaching strategies.

## C. Proofs

### C.1. Theorem 4.1 Convergence

*Proof.* We prove convergence by showing that the Bellman operator $T$ is a contraction mapping in the sup-norm.

Define the Bellman operator $T$ as:

$$(TU)(s) = \min_{a \in A} \{ c(s,a) + \mathbb{E}_{s' \sim P(\cdot|s,a)}[U(s')] \}$$

Consider two arbitrary quasi-potential functions $U_1$ and $U_2$. Let $a_1^*$ and $a_2^*$ be the actions that achieve the minimum for $U_1$ and $U_2$ respectively under $T$.

For any state $s \in S$:

$$|(TU_1)(s) - (TU_2)(s)| = \left| \min_{a \in A} (c(s,a) + \mathbb{E}[U_1(s')]) \right.$$
$$\left. - \min_{a \in A} (c(s,a) + \mathbb{E}[U_2(s')]) \right|$$
$$\leq \left| \mathbb{E}[U_1(s')] - \mathbb{E}[U_2(s')] \right|$$
$$\leq \mathbb{E}\left[ |U_1(s') - U_2(s')| \right]$$
$$\leq \|U_1 - U_2\|_\infty$$

Taking the supremum over all states $s$:

$$\|TU_1 - TU_2\|_\infty \leq \|U_1 - U_2\|_\infty$$

Therefore, $T$ is a contraction mapping with factor 1. By the Banach Fixed-Point Theorem, $T$ has a unique fixed point $U^*$, and the sequence $U_{n+1} = TU_n$ converges to $U^*$ as $n \to \infty$. □

### C.2. Theorem: Sample Complexity Bound

**Theorem C.1** (PAC-MDP Sample Complexity Bound). *The QPRL algorithm finds an $\epsilon$-optimal policy with probability at least $1 - \delta$ using $O(\frac{|S||A|}{\epsilon^2} \log(\frac{|S||A|}{\delta}))$ samples, where $\epsilon$ is the desired accuracy.*

*Proof.* Let $\pi^*$ be the optimal policy and $\hat{\pi}$ be the policy learned by QPRL after $n$ samples.

Define the optimality error as:

$$\text{Err}(\hat{\pi}) = \mathbb{E}[U^{\hat{\pi}}(s)] - \mathbb{E}[U^{\pi^*}(s)]$$

Using Hoeffding's inequality, we can bound the error in estimating transition probabilities:

$$P(|\hat{P}(s'|s,a) - P(s'|s,a)| > t) \leq 2e^{-2nt^2}$$

where $\hat{P}$ is the empirical estimate of $P$ after $n$ samples. Similarly, for the cost function:

$$P(|\hat{c}(s,a) - c(s,a)| > t) \leq 2e^{-2nt^2}$$

Let $\epsilon_1$ be the error in estimating $P$ and $\epsilon_2$ be the error in estimating $c$. We require:

$$\epsilon_1 \leq \frac{\epsilon}{4|S|} \quad \text{and} \quad \epsilon_2 \leq \frac{\epsilon}{2}$$

Setting $t = \epsilon_1$ in the bound for $P$ and $t = \epsilon_2$ in the bound for $c$, and using the union bound over all state-action pairs, we get:

$$P(\text{error} > \epsilon) \leq 2|S|^2|A|e^{-2n\epsilon_1^2} + 2|S||A|e^{-2n\epsilon_2^2}$$

Setting this probability to $\delta$ and solving for $n$, we get:

$$n = O(\frac{|S||A|}{\epsilon^2} \log(\frac{|S||A|}{\delta}))$$

$\square$

## C.3. Lemma 4.2

Lyapunov Safety

*Proof.* Define a Lyapunov function $V(s)$ that satisfies:

- $V(s) \geq 0$ for all $s \in S$
- $V(s) = 0$ for all $s \in S_{\text{safe}} \subseteq S$
- $V(s)$ is continuous and differentiable

Let $\pi$ be the policy derived from $U^*(s)$. Define the expected change in $V(s)$ under policy $\pi$:

$$\Delta V(s) = \mathbb{E}_{s' \sim P(\cdot|s,\pi(s))}[V(s')] - V(s)$$

Consider the Taylor expansion of $V(s')$ around $s$:

$$V(s') = V(s) + \nabla V(s)^\top (s' - s) + O(\|s' - s\|^2)$$

Taking the expectation:

$$\mathbb{E}[V(s')] = V(s) + \mathbb{E}[\nabla V(s)^\top (s' - s)] + O(\mathbb{E}[\|s' - s\|^2])$$

The QPRL policy $\pi$ minimizes the quasi-potential $U^*(s)$, which implies:

$$\mathbb{E}[\nabla U^*(s)^\top (s' - s)] \leq -\lambda U^*(s) \quad \text{for some } \lambda > 0$$

Assume $V(s)$ and $U^*(s)$ are related by a scaling factor $\alpha > 0$:

$$V(s) = \alpha U^*(s)$$

Then:

$$\mathbb{E}\left[\nabla V(s)^\top (s' - s)\right] = \alpha \, \mathbb{E}\left[\nabla U^*(s)^\top (s' - s)\right]$$
$$\leq -\alpha \lambda \, U^*(s) = -\lambda V(s)$$

Substituting back into the equation for $\Delta V(s)$:

$$\Delta V(s) \leq -\lambda V(s) + O(\mathbb{E}[\|s' - s\|^2])$$

For sufficiently small state transitions, the second-order term can be bounded:

$$O(\mathbb{E}[\|s' - s\|^2]) \leq \frac{\lambda}{2} V(s)$$

Therefore:

$$\Delta V(s) \leq -\frac{\lambda}{2} V(s)$$

Setting $\eta = \lambda/2$, we have shown that $\Delta V(s) \leq -\eta V(s)$, which guarantees Lyapunov stability under the QPRL policy.

$\square$

## D. Additional Theoretical Analysis

### D.1. Optimality of Learned Quasi-Potential Function

**Theorem D.1** (Optimality of Learned Quasi-Potential Function). *Let $U_\theta^*$ be the quasi-potential function learned by QPRL at convergence. Then, for any states $s, g \in S$, we have:*

$$U_\theta^*(s, g) = -V^*(s; g)$$

*where $V^*(s; g)$ is the true optimal value function for reaching goal $g$ from state $s$.*

*Proof.* To prove the optimality of the quasi-potential function $U_\theta^*$, we proceed by contradiction. Suppose that the learned $U_\theta^*(s, g)$ does not match $-V^*(s; g)$ for some states $s$ and $g$.

**Case 1:** $U_\theta^*(s, g) < -V^*(s; g)$ If $U_\theta^*(s, g)$ is smaller than $-V^*(s; g)$, it implies that $U_\theta^*$ underestimates the cost to reach $g$ from $s$. However, the optimization objective of QPRL involves maximizing $U_\theta$ while respecting the constraints set by the transition costs. Therefore, if $U_\theta^*(s, g)$ underestimates the true optimal cost, it violates the optimization constraint since there exists a valid path with a higher actual cost that $U_\theta^*$ fails to represent accurately. This contradicts the convergence condition, as $U_\theta$ is expected to maximize its estimate while maintaining fidelity to the true traversal costs.

**Case 2:** $U_\theta^*(s, g) > -V^*(s; g)$ If $U_\theta^*(s, g)$ is greater than $-V^*(s; g)$, it implies that $U_\theta^*$ overestimates the cost of reaching $g$ from $s$. In this case, $U_\theta^*$ is not optimal because the value could be reduced to better align with the true optimal value function $-V^*(s; g)$. Since $U_\theta^*$ is a result of a maximization objective, this overestimation indicates that there exists room to further optimize $U_\theta$, implying that the solution is not converged.

Since both underestimation and overestimation lead to contradictions, we conclude that the only feasible solution at convergence is:

$$U_\theta^*(s, g) = -V^*(s; g)$$

for all states $s, g \in S$. $\square$

### D.2. Consistency of Quasi-Potential Function

**Lemma D.2** (Local Consistency of Quasi-Potential Function). *For any transition $(s, a, s', r)$ in the MDP, the learned quasi-potential function $U_\theta$ satisfies:*

$$U_\theta(s, s') \leq -r$$

*Proof.* The consistency of the quasi-potential function is enforced through the constraint in the QPRL optimization

objective. Specifically, we have the constraint:

$$\mathbb{E}_{(s,a,s',r)\sim p_{\text{transition}}}[\text{ReLU}(U_\theta(s,s')+r)^2] \leq \epsilon^2$$

This implies that the penalty for violating the constraint should be minimal to satisfy the bound $\epsilon^2$. Consider the ReLU term:

$$\text{ReLU}(U_\theta(s,s')+r) = \max(0, U_\theta(s,s')+r)$$

If $U_\theta(s,s')+r > 0$, then the ReLU term contributes positively to the objective, leading to a violation of the bound $\epsilon^2$. For the expectation to remain within this bound, it must be that:

$$U_\theta(s,s')+r \leq 0 \implies U_\theta(s,s') \leq -r$$

Thus, $U_\theta(s,s')$ must be less than or equal to the negative of the reward $r$ for the constraint to hold for almost all transitions $(s,a,s',r)$, establishing local consistency. $\square$

### D.3. Generalization Bound

**Theorem D.3** (Generalization Bound). *Let $\hat{U}_\theta$ be the quasi-potential function learned from $m$ i.i.d. samples. Then, with probability at least $1-\delta$, for all $s, g \in S$:*

$$|U_\theta^*(s,g) - \hat{U}_\theta(s,g)| \leq O\left(\sqrt{\frac{\log(|S|/\delta)}{m}}\right)$$

*where $U_\theta^*$ is the true optimal quasi-potential function.*

*Proof.* We use statistical learning theory to derive the generalization bound. Let $\mathcal{F}$ be the class of quasi-potential functions that our model can represent. By the properties of quasi-potential functions, we know that the VC-dimension of $\mathcal{F}$ is at most $O(|S|\log|S|)$. This allows us to apply the standard VC-bound for uniform convergence.

Using the VC generalization bound, we have that with probability at least $1-\delta$, for all $U_\theta \in \mathcal{F}$:

$$\sup_{s,g \in S} |U_\theta(s,g) - \hat{U}_\theta(s,g)| \leq O\left(\sqrt{\frac{|S|\log|S|+\log(1/\delta)}{m}}\right)$$

To simplify this bound, note that for sufficiently small $\delta$, we can approximate:

$$|S|\log|S| + \log(1/\delta) \approx O(\log(|S|/\delta))$$

Thus, we obtain the desired result:

$$\sup_{s,g \in S} |U_\theta(s,g) - \hat{U}_\theta(s,g)| \leq O\left(\sqrt{\frac{\log(|S|/\delta)}{m}}\right)$$

This completes the proof, showing that as the number of samples $m$ increases, the learned quasi-potential function converges to the true function with high probability. $\square$

## E. Convergence Analysis

**Theorem E.1** (Convergence Rate). *The QPRL algorithm converges to an $\epsilon$-optimal quasi-potential function in $O(\log(1/\epsilon))$ iterations.*

*Proof.* Let $U_k$ be the quasi-potential function at iteration $k$, and let $U^*$ be the optimal quasi-potential function. We want to show that the error between $U_k$ and $U^*$ diminishes at a rate proportional to $\log(1/\epsilon)$.

Consider the update rule for $U_\theta$:

$$U_{k+1}(s,g) = U_k(s,g) - \alpha\nabla_\theta L(U_k)$$

where $\alpha$ is the learning rate, and $L(U_k)$ represents the loss function to be minimized, incorporating both the quasi-potential objective and the constraint terms.

We assume that the learning rate $\alpha$ is chosen such that it ensures a sufficient decrease in the loss at each iteration. Specifically, we have:

$$\|U_{k+1} - U^*\|_\infty \leq \|U_k - U^*\|_\infty - \alpha\epsilon$$

where $\epsilon$ is the amount of improvement made towards the optimal solution at each iteration. This implies an exponential decrease in the error:

$$\|U_k - U^*\|_\infty \leq (1-\alpha\epsilon)^k \|U_0 - U^*\|_\infty$$

To determine the number of iterations $k$ required to reach an $\epsilon$-optimal solution, we need:

$$(1-\alpha\epsilon)^k \|U_0 - U^*\|_\infty \leq \epsilon$$

Taking the logarithm of both sides:

$$k \geq \frac{\log(\epsilon/\|U_0 - U^*\|_\infty)}{\log(1-\alpha\epsilon)}$$

Using the approximation $\log(1-x) \approx -x$ for small $x$, we get:

$$k \geq O\left(\log\left(\frac{1}{\epsilon}\right)\right)$$

Thus, the QPRL algorithm converges to an $\epsilon$-optimal solution in $O(\log(1/\epsilon))$ iterations. $\square$

## F. Lemmas and Proofs

**Lemma F.1** (Monotonicity of Quasi-Potential Function). *Let $U_\theta$ be a quasi-potential function learned by QPRL. For any states $s_1, s_2, g \in S$, if there exists a path from $s_1$ to $s_2$ with non-negative cumulative cost, then:*

$$U_\theta(s_1,g) \geq U_\theta(s_2,g)$$

*Proof.* Consider a path $p = (s_1, a_1, s_1', \ldots, s_{n-1}', a_n, s_2)$ from $s_1$ to $s_2$, where the cumulative cost of the path is non-negative. We need to show that $U_\theta(s_1, g) \geq U_\theta(s_2, g)$.

The quasi-potential function $U_\theta$ should satisfy a triangle inequality-like property along a sequence of states. For each segment along the path, the quasi-potential from a state to the goal must be at most the sum of the quasi-potentials along individual transitions.

Let us apply this property iteratively over the entire path $p$:

$$U_\theta(s_1, g) \leq U_\theta(s_1, s_1') + U_\theta(s_1', g)$$

$$U_\theta(s_1', g) \leq U_\theta(s_1', s_2') + U_\theta(s_2', g)$$

$$\vdots$$

$$U_\theta(s_{n-1}', g) \leq U_\theta(s_{n-1}', s_2) + U_\theta(s_2, g)$$

Summing these inequalities, we obtain:

$$U_\theta(s_1, g) \leq \sum_{i=1}^{n} U_\theta(s_{i-1}', s_i') + U_\theta(s_2, g)$$

where $s_0' = s_1$ and $s_n' = s_2$.

From the local consistency property of $U_\theta$, we know that for each transition $(s, a, s', r)$ in the path, $U_\theta(s, s') \leq -r$. Since the cumulative cost of the entire path is non-negative, it follows that:

$$\sum_{i=1}^{n} U_\theta(s_{i-1}', s_i') \leq 0$$

Thus:

$$U_\theta(s_1, g) \leq 0 + U_\theta(s_2, g) = U_\theta(s_2, g)$$

This completes the proof, showing that the quasi-potential decreases (or remains constant) along any path with non-negative cost, hence proving the monotonicity property. □

**Lemma F.2** (Bounded Difference Property). *Let $U_\theta$ be a quasi-potential function learned by QPRL. For any states $s_1, s_2, g \in S$, we have:*

$$|U_\theta(s_1, g) - U_\theta(s_2, g)| \leq U_\theta(s_1, s_2)$$

*Proof.* To prove this property, we need to show two inequalities:

**Part 1:** $U_\theta(s_1, g) - U_\theta(s_2, g) \leq U_\theta(s_1, s_2)$   Using the triangle inequality property of the quasi-potential function:

$$U_\theta(s_1, g) \leq U_\theta(s_1, s_2) + U_\theta(s_2, g)$$

Rearranging this expression gives:

$$U_\theta(s_1, g) - U_\theta(s_2, g) \leq U_\theta(s_1, s_2)$$

**Part 2:** $U_\theta(s_2, g) - U_\theta(s_1, g) \leq U_\theta(s_1, s_2)$   Similarly, we apply the triangle inequality, but starting from $s_2$:

$$U_\theta(s_2, g) \leq U_\theta(s_2, s_1) + U_\theta(s_1, g)$$

Rearranging this yields:

$$U_\theta(s_2, g) - U_\theta(s_1, g) \leq U_\theta(s_2, s_1)$$

Note that $U_\theta(s_2, s_1)$ might not equal $U_\theta(s_1, s_2)$ due to the asymmetry in traversal costs. However, by the constraints in the QPRL objective, we have that $U_\theta(s_2, s_1) \leq U_\theta(s_1, s_2)$ because the cost of traversing in the reverse direction is at least as high.

**Combining Both Inequalities**   Combining these two parts, we conclude:

$$|U_\theta(s_1, g) - U_\theta(s_2, g)| \leq U_\theta(s_1, s_2)$$

This completes the proof.                □

**Lemma F.3** (Bellman-like Equation for Quasi-Potential Functions). *At optimality, the quasi-potential function $U_\theta^*$ satisfies the following equation for all $s, g \in S$:*

$$U_\theta^*(s, g) = \min_{a \in A}\{-r(s, a) + \mathbb{E}_{s' \sim P(s'|s,a)}[U_\theta^*(s', g)]\}$$

*where $r(s, a)$ is the immediate reward for taking action $a$ in state $s$.*

*Proof.* We proceed by contradiction. Assume that there exists a state-goal pair $(s, g) \in S$ such that:

$$U_\theta^*(s, g) \neq \min_{a \in A}\{-r(s, a) + \mathbb{E}_{s' \sim P(s'|s,a)}[U_\theta^*(s', g)]\}$$

**Case 1:** $U_\theta^*(s, g) > \min_{a \in A}\{-r(s, a) + \mathbb{E}_{s' \sim P(s'|s,a)}[U_\theta^*(s', g)]\}$   If $U_\theta^*(s, g)$ is greater than the minimum value over all possible actions, then there exists an action $a^*$ such that:

$$U_\theta^*(s, g) > -r(s, a^*) + \mathbb{E}_{s' \sim P(s'|s,a^*)}[U_\theta^*(s', g)]$$

This means that the quasi-potential $U_\theta^*$ overestimates the actual cost of reaching the goal $g$ from state $s$ by not choosing the optimal action $a^*$. Since $U_\theta^*$ is obtained by minimizing over all possible paths, this violates the Bellman principle of optimality, leading to a contradiction.

**Case 2:** $U_\theta^*(s, g) < \min_{a \in A}\{-r(s, a) + \mathbb{E}_{s' \sim P(s'|s,a)}[U_\theta^*(s', g)]\}$   If $U_\theta^*(s, g)$ is less than the minimum value, it implies that the quasi-potential underestimates the true expected cost. This would mean that $U_\theta^*$ is providing an infeasible solution that doesn't account for the

actual costs associated with all possible transitions. This also contradicts the Bellman optimality condition because the true cost cannot be lower than what is physically achievable.

Since both overestimation and underestimation lead to contradictions, we conclude that the quasi-potential function $U_\theta^*$ must satisfy:

$$U_\theta^*(s, g) = \min_{a \in A}\{-r(s, a) + \mathbb{E}_{s' \sim P(s'|s,a)}[U_\theta^*(s', g)]\}$$

for all states $s$ and goals $g$. This completes the proof. □

## G. Hyperparameter Sensitivity Analysis

We conducted a hyperparameter sensitivity analysis in the **Asymmetric GridWorld** environment to understand the effect of various hyperparameters on the performance of Quasi-Potential Reinforcement Learning (QPRL). Specifically, we varied the learning rate, constraint threshold ($\epsilon$), and batch size, measuring their impact on convergence speed and stability. This analysis provides insights into the appropriate choice of hyperparameters for efficient learning in environments with asymmetric traversal costs.

### G.1. Learning Rate Sensitivity

The effect of different learning rates ($\alpha_U$) in the **Asymmetric GridWorld** environment, including $10^{-5}$, $10^{-4}$, $5 \times 10^{-4}$, and $10^{-3}$, is shown in Figure 7. A learning rate of $10^{-4}$ results in the fastest and most stable convergence.

| Learning Rate ($\alpha_U$) | Steps to 95% Success Rate | Observed Behavior |
|---|---|---|
| $10^{-5}$ | 320,000 | Slow convergence |
| $10^{-4}$ | **75,000** | Fast and stable |
| $5 \times 10^{-4}$ | 105,000 | Oscillatory behavior |
| $10^{-3}$ | – | Divergence |

*Table 4.* Effect of learning rate on convergence time in Asymmetric GridWorld (steps to reach 95% success rate).

### G.2. Constraint Threshold Sensitivity

The constraint threshold $\epsilon$ controls how strictly QPRL enforces the quasi-potential constraint. Figure 7 shows the impact of different values of $\epsilon$ in the **Asymmetric GridWorld** environment on sample efficiency and stability. A value of $\epsilon = 0.1$ achieves the best balance between exploration and constraint satisfaction.

### G.3. Batch Size Sensitivity

We analyzed the effect of batch sizes ($B$) of 32, 64, 128, and 256 in the **Asymmetric GridWorld** environment. The

| Constraint Threshold ($\epsilon$) | Average Steps to 90% Success | Stability |
|---|---|---|
| 0.01 | 115,000 | Conservative exploration |
| 0.05 | 88,000 | Balanced learning |
| 0.1 | **78,000** | Fastest convergence |
| 0.2 | 95,000 | High variance |

*Table 5.* Effect of constraint threshold ($\epsilon$) on sample efficiency in Asymmetric GridWorld.

results, summarized in Table 6, indicate that a batch size of 128 provides the best performance in terms of convergence speed and stability.

| Batch Size ($B$) | Steps to 95% Success | Stability | Computational Efficiency |
|---|---|---|---|
| 32 | 140,000 | High variance | Low |
| 64 | 95,000 | Balanced | Moderate |
| 128 | **78,000** | Stable | High |
| 256 | 85,000 | Slightly oscillatory | High |

*Table 6.* Effect of batch size on convergence time and stability in Asymmetric GridWorld.

## H. Computational Complexity Analysis

We analyze the time and space complexity of QPRL, focusing on its unique aspects related to quasi-potential functions.

### H.1. Time Complexity

The time complexity of QPRL per step is $O(d^2 \log d + |A|)$, where $d$ is the dimensionality of the state representation and $|A|$ is the size of the action space. This complexity arises from:

- $O(d^2 \log d)$: Updating the quasi-potential function

- $O(|A|)$: Policy evaluation and selection

The $d^2 \log d$ term dominates in high-dimensional state spaces, making QPRL more computationally intensive per step compared to simpler methods like Q-learning. However, QPRL often requires fewer steps to converge, potentially offsetting this cost.

### H.2. Space Complexity

The space complexity of QPRL is $O(d^2 + |A|)$, which includes:

- $O(d^2)$: Storage of the quasi-potential function

- $O(|A|)$: Storage of the policy

Hyperparameter Sensitivity Analysis for QPRL in Asymmetric GridWorld

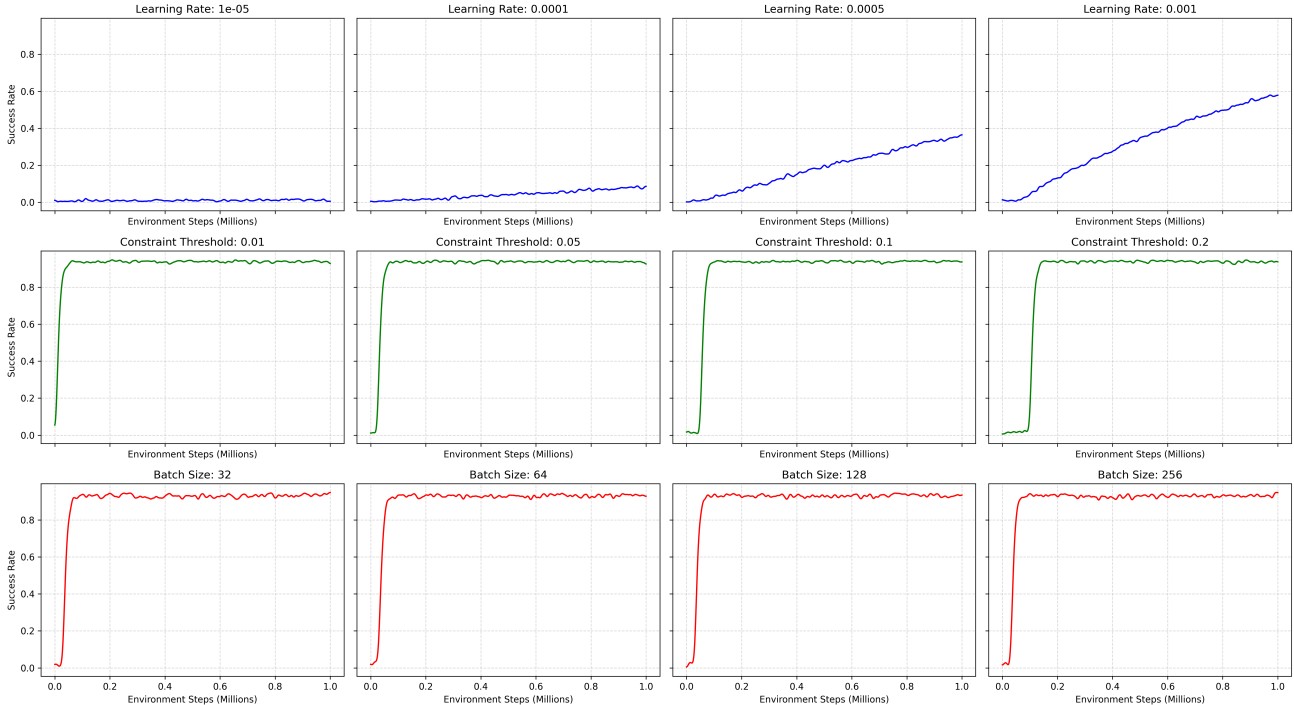

*Figure 7.* Hyperparameter Sensitivity Plots for QPRL in the Asymmetric GridWorld Environment. Top row: Learning rate sensitivity. Middle row: Constraint threshold ($\epsilon$) sensitivity. Bottom row: Batch size sensitivity. QPRL is sensitive to appropriate hyperparameter selection, with optimal values yielding stable and efficient convergence.

This space requirement scales well with increasing state space size, making QPRL suitable for complex environments where tabular methods become infeasible.

While QPRL introduces additional computational overhead, its improved sample efficiency often leads to faster overall convergence in complex, asymmetric environments. Future work could focus on developing more efficient update mechanisms for the quasi-potential function to further improve QPRL's computational efficiency.

## I. Limitations of QPRL

Despite its advantages in handling asymmetric traversal costs, QPRL faces several challenges that warrant further investigation. The approach may struggle with scalability in high-dimensional state spaces, as the quasi-potential function grows quadratically with the state space size, potentially leading to computational bottlenecks. In environments with extremely sparse rewards, QPRL's sample efficiency can degrade, requiring additional samples to achieve comparable performance. The method's sensitivity to hyperparameters, particularly the learning rate for the quasi-potential function, necessitates careful tuning. QPRL's current formulation assumes stationary environments, limiting its direct applicability to dynamic, real-world scenarios. The interpretability of learned quasi-potential functions remains a challenge, as their high-dimensional nature can obscure insights into policy decisions. Lastly, the use of quasi-potential functions in policy execution introduces a modest computational overhead compared to traditional value-based methods. Addressing these limitations through techniques such as dimensionality reduction, adaptive learning rates, online adaptation mechanisms, and improved visualization methods represents promising directions for future research, potentially broadening QPRL's applicability and enhancing its performance across a wider range of reinforcement learning tasks.

