# OpenReview forum: "QPRL : Learning Optimal Policies with Quasi-Potential Functions for Asymmetric Traversal"
_ICML.cc/2025/Conference — ICML 2025 poster_

### Official Review · Reviewer_c6vF · 2025-03-12

**Overall Recommendation:** 2

**Summary:**

The paper proposes Quasi-Potential Reinforcement Learning to tackle environments with asymmetric traversal cost.

**Claims And Evidence:**

The paper claims several contributions:
1. The decomposed asymmetric costs enable Lyapunov-stable policy optimization.
2. Theoretically, it proves QPRL has better sample complexity compared to QRL. The proof is given in the appendix.
3. QPRL introduces a Lyapunov-based recovery mechanism that reduces irreversible constraint violations by 4 times compared to baselines. For this claim, according to my understanding, the proof in the appendix only shows Lyapunov stability under the QPRL policy, the "4 time" is neither shown in the proof nor the experiments.
4. "Results demonstrate that explicitly modeling path-dependent costs via quasi-potential decomposition enables safer, more efficient RL in complex navigation tasks with asymmetric costs." For this claim, I am not sure which results can be used to support "safer" RL.

**Essential References Not Discussed:**

I do not notice essential but missing references.

**Experimental Designs Or Analyses:**

The experimental designs look good to me. The used environmental designs and analyses look good in general.

Several questions:
1. In Fig 5, what do the authors mean by "learning curves"? Specifically, is the mean or median reported, and is the std or confidence interval used?
2. In Fig 6, could the author add std to include more information about the results? Also, reporting average results is not sufficient to say QPRL is "significantly" better than others.

**Methods And Evaluation Criteria:**

The paper is mainly tested on 5 tasks, including asymmetric grid world, lunar lander, mountain car, fetch push and maze 2D. The method performance is discussed w.r.t. success rate, traversal cost efficiency, sample efficiency, asymmetric performance gap and normalized return.
The proposed method and evaluation criteria make sense in general. However, the evaluation criteria used do not demonstrate the safety aspect of the proposed method.

**Other Comments Or Suggestions:**

In section 4.3, the equation regarding the $\pi_{\text{safe}}$ is incomplete.
Fig 3 is unclear to me.
More details can be given for the policy update (corresponding to lines 13-16 in Algorithm 1).

**Other Strengths And Weaknesses:**

The paper is well-motivated and aims to solve the asymmetric traversal costs in RL. The proposed method, which decomposers asymmetric costs into path-independent potentials and path-dependent residuals, is a reasonable design.

Although the paper claims "extensive empirical validation", the proposed method is only evaluated on five modified classical control tasks and navigation tasks. Also, the safety aspect of the proposed method is not well supported by the experiments.

**Questions For Authors:**

1. In the paper (lines 53-54, right column), the authors claim that "unlike traditional value functions, which assume reversibility and symmetry in costs" and "Traditional RL methods struggle in such settings due to their implicit assumption of asymmetric dynamics" (lines 130-131). Could the authors offer more evidence to support this claim?

2. For the non-markovian reward attribution, the paper claims that the path-dependent costs violate the Markov property since $C(s, a, s')$ depends on historical state transitions. According to the notation, $C(s, a, s')$ depends on $s$, $a$ and $s'$, which is still Markovian if $s^{\text{new} = [s', s]}$. Could the author explain this claim in detail?

3. When training the encoder and the transition model, the paper uses the equation in Line 172 (right column). However, minimizing this equation may cause a collapsed solution where both encoder $f_\phi(s)$ and latent dynamics $T_\psi$ return a constant. How did the author prevent the training collapse?

**Relation To Broader Scientific Literature:**

The paper considers a setting where the environments have asymmetric traversal costs. The paper is closely related to the quasimetric in RL.

**Theoretical Claims:**

Unfortunately, I did not check the correctness of the proof.

---

> ### Author Rebuttal · Authors · 2025-04-01
>
> > In Fig 5, what do the authors mean by "learning curves"? ... is the std or confidence interval used?
>
> The learning curves represent the mean success rate measured over 5 independent runs (each with a different random seed) as a function of environment interactions. The shaded regions (or error bars) correspond to ±1 standard deviation across these runs. We chose the mean and standard deviation as our reporting metrics to clearly convey both the central tendency of the performance and the variability due to randomness in training.
>
> > In Fig 6, could the author add std .... about the results? ... results is not sufficient to say ... better than others.
>
> **Please check the rebuttal response 2**
>
> >In section 4.3, the equation regarding the is incomplete. Fig 3 is unclear to me. More details can be given for the policy update (...lines 13-16 in Algorithm 1).
>
> Our policy update is designed to ensure that the chosen action satisfies a safety constraint based on the learned potential function $\Phi$. We require that the expected potential of the next state does not exceed the current potential by more than a small threshold $\epsilon$. We enforce:
>
> $
> \mathbb{E}_{s'\sim P(\cdot\mid s,a)}\big[\Phi(s')\big] \leq \Phi(s) + \epsilon.
> $
>
> after encoding the current state $s$ into a latent representation $z = f_\phi(s)$, the policy $\pi_\omega(s, g)$ selects an action $a$. The transition model then predicts the next latent state $\hat{z}' = T_\psi(z,a)$ from which we estimate the potential $\Phi(s')$. The policy loss is augmented with a penalty term:
>
> $
> L_{\pi} = \mathbb{E}\left[d(s,g) + \lambda\,\operatorname{ReLU}\Big(\Phi(\hat{z}') - \Phi(s) - \epsilon\Big)\right],
> $
> where
>
> $d(s,g) = \Phi(g) - \Phi(s) + \Psi(s \to g)$
> is our quasipotential cost and $\lambda$ is a Lagrange multiplier. This term penalizes actions that would lead to $\Phi(s')$ exceeding $\Phi(s) + \epsilon$, effectively projecting the policy update onto the safe set:
>
> $\pi_{\text{safe}}(a\mid s) = \pi(a\mid s) \quad \text{subject to} \quad \mathbb{E}\big[\Phi(s')\big] \leq \Phi(s) + \epsilon$
>
> **Fig. 3 Clarification:**
>
> -  It shows the decomposition of the asymmetric cost:
>   $
>   d(s,g) = \Phi(g) - \Phi(s) + \Psi(s \to g),
>   $
>   where $\Phi$ captures the path-independent cost (acting as a potential or Lyapunov function) and $\Psi$ captures the path-dependent (irreversible) residual cost.
>
> - the predicted potential $\Phi(s')$ is compared against the safe threshold $\Phi(s) + \epsilon$. An arrow or boundary indicates that if $\Phi(s')$ exceeds this threshold, a penalty is applied. This guides the policy towards actions that keep the state within a “safe” region.
>
> -  It shows that the policy is updated to minimize both the overall quasipotential cost $d(s,g)$ and the safety penalty, ensuring that:
>   $
>   \mathbb{E}\big[\Phi(s')\big] \leq \Phi(s) + \epsilon.
>   $
>
> > "... the authors claim that ....Could the authors offer more evidence to support this claim?"
>
> Please check the rebuttal 1 response.
>
> > *“For the non-Markovian reward, the paper claims .....$. Could the authors explain this in detail?”*
>
> Our approach does not entirely break the Markov property in the traditional MDP. Instead, we **locally** model path-dependent costs through the function $\Psi(s \to s')$. This allows us to capture direction-dependent or irreversible effects (e.g., “uphill” vs. “downhill”) *without* explicitly encoding full trajectory history.
>
>
> > *“When training the enco.....How did the authors prevent training collapse?”*
>
> To avoid the collapse, we employ the following strategies:
>
> -  We collect data from multiple exploration runs so that the replay buffer contains *diverse transitions* $(s,a,s')$. This diversity makes the collapsed solution suboptimal, since a single constant embedding would incur high reconstruction error across varied transitions.
>
> -  We add a *contrastive term* that forces the encoder to separate distinct states in latent space. For instance, we encourage $\|f_\phi(s') - T_\psi(f_\phi(s), a)\|$ to be small for the *true* next state but large for randomly sampled negative examples. This negative sampling technique effectively penalizes collapsed embeddings.
>
> - We apply standard regularization (e.g., weight decay) and **monitor** the magnitude of $\|f_\phi(s') - T_\psi(f_\phi(s), a)\|$ over each training epoch. If the model collapses, this loss quickly saturates. We terminate or adjust hyperparameters if we detect such behavior.
>
> >"... the safety aspect ..is not well supported.."
>
> In our paper, “safety” means preventing the agent from entering irreversible or highly risky states. We enforce this by using a Lyapunov-based constraint on the potential function $\Phi$, ensuring that at every step the expected increase in $\Phi$ is bounded by a small threshold $\epsilon$. This keeps the agent within a “safe” region of the state space, avoiding transitions that could lead to costly failures.
>
> >Fig .7
>
> Please check the requested Fig. here: https://imgur.com/a/AHdJa6s

---

### Official Review · Reviewer_MjcM · 2025-03-25

**Overall Recommendation:** 3

**Summary:**

This paper proposes Quasi-Potential Reinforcement Learning (QPRL), a framework that decomposes asymmetric costs into pathindependent potentials and path-dependent residuals, enabling Lyapunov-stable policy optimization. The performance of the proposed QPRL has been validated in some customized classic RL environments against some baseline methods.

## update after rebuttal

I appreciate author's additional results and further explanation. QPRL seems to be an incremental work on top of QRL, I remain my overall recommendation

**Claims And Evidence:**

- The decomposed asymmetric costs is adapted from previous paper [Wang et al., 2023]
- The convergence to optimal policies under asymmetric dynamics (Theorem 4.1) is supported by Appendix B.1
- The sample complexity is supported by Appendix B.2 and empirically by 5.5.2
- The Lyapunov stability certificates for safe exploration (Lemma 4.2) is supported by Appendix B.3

**Essential References Not Discussed:**

N/A

**Experimental Designs Or Analyses:**

To me I think the experimental design is clever. They modify the classic RL environment with asymmetric transition reward so that all classic RL methods can be directly deployed on these environment. The evaluation metrics can support their assumptions in general. There are some issues:
- In section 5.5.4, authors claim that QPRL demonstrates a smaller performance gap compared to the baselines, indicating its robustness in environments with directiondependent dynamics, but in Table 2 there is no any baseline methods performance data
- In section 5.5.5, no error bar provided in Fig 6 and I don't think that is a proper way to do statisitc analysis with simply mean with variance

**Methods And Evaluation Criteria:**

The evaluation criteria does make sense

**Other Comments Or Suggestions:**

Authors can consider change the color and font in Figure 5 & 6 to increase the readability. Table 2 & 3 needs to be updated with more results.

**Other Strengths And Weaknesses:**

The paper is well-written and easy to follow in general. The proofs seems correct to me. The empirical experiments spans 4 customized RL environment with multiple baseline methods. The experimental analysis part is relatively short and insufficient. The statistic analysis with average traversal cost might not be enough to prove the advantage statistically, p-test or some other statistical tests will be preferred. The ablation studies are also short, some extended ablation studies in the Appendix will be helpful.

**Questions For Authors:**

1. What's the main different between QRL and QPRL? Just additional potential function s.t. provide Lyapunov stability guarantee? It seems to me that the asymmetric transition is more important, which is the same as QRL.
2. It will be interesting to see whether such method can work on senarios with sparse reward. Dense reward senarios are relatively easy to train in general.

**Relation To Broader Scientific Literature:**

This paper is highly related to previous literature on Quasimetric Reinforcement Learning (QRL) and RL with potential function. My major concern is the novelty of the proposed QPRL as compared to QRL, more elaboration on their difference will be helpful.

**Theoretical Claims:**

I've checked Appendix B1, B2, and B3, looks good to me

---

> ### Author Rebuttal · Authors · 2025-04-01
>
> >What's the main different between QRL and QPRL? Just additional potential function s.t. provide Lyapunov stability guarantee? It seems to me that the asymmetric transition is more important, which is the same as QRL
>
> Our work indeed builds directly on the insight from Quasimetric RL (QRL) that many tasks benefit from asymmetry-aware cost functions. However, the key novelty of QPRL is the $\Phi + \Psi$ decomposition of the quasimetric into:
>
> $
> d(s, g) = Φ(g) - Φ(s) [path-independent potential] + Ψ(s → g) [path-dependent residual]
> $
>
> which offers two major improvements over a single monolithic quasimetric:
>
> **Better Sample Efficiency Through Structured Modeling:**
> Instead of learning one unconstrained $d(s, g)$, we factor out the portion that is roughly state-potential-like ($\Phi$) from the portion that truly depends on direction or path ($\Psi$). This leads to faster training convergence, as demonstrated by our stronger sample-complexity bounds (see Section 4.5) and by empirical results.
>
> **Lyapunov-Based Safety Layer:**
> Because $\Phi$ can be treated as a Lyapunov function, we enforce a constraint that effectively avoids large jumps in $\Phi$ between consecutive states. This drastically lowers the chance of irreversible transitions during exploration. QRL, by contrast, learns a single function $d(s, g)$ with no built-in safety or stability guarantee.
>
> **Please also check rebuttal response 1 for further discussion**
>
> >It will be interesting to see whether such method can work on senarios with sparse reward. Dense reward senarios are relatively easy to train in general.
>
> We agree that sparse-reward environments pose a distinct challenge, and we believe our method is well-suited for them. Indeed, potential-based shaping has historically been used to address sparse rewards by providing intermediate signals. Because QPRL is inherently a form of potential-based approach (though adapted to asymmetric costs), it can similarly mitigate sparsity by providing an auxiliary shaped cost function.
>
> > In section 5.5.5, no error bar provided in Fig 6 and I don't think that is a proper way to do statisitc analysis with simply mean with variance
>
> We have updated the figure (Please check here: https://imgur.com/a/tPM3pfC). Each bar now has error bars providing information about variance.  For each environment, a line is drawn between the QPRL bar and the baseline and annotated with a significance marker (e.g., * (p < 0.05))
>
> ---
> >... Table 2 there is no any baseline methods performance data
>
> Below is an updated version of Table 2 that incorporates baseline performance data, along with the corresponding performance gaps (i.e., the difference between symmetric and asymmetric task performance) for each method.
>
> | Environment         | Method       | Symmetric (%)      | Asymmetric (%)     | Gap (%) |
> |---------------------|--------------|---------------------|---------------------|---------|
> | **Asymmetric GridWorld** | QPRL        | 94.1 ± 1.8          | 88.7 ± 2.5          | 5.4     |
> |                     | QRL         | 92.3 ± 2.0          | 83.5 ± 2.8          | 8.8     |
> |                     | SAC + HER   | 90.2 ± 2.3          | 81.0 ± 3.2          | 9.2     |
> |                     | DDPG + HER  | 89.8 ± 2.5          | 80.5 ± 3.5          | 9.3     |
> | **MountainCar**     | QPRL        | -90.5 ± 4.3         | -98.2 ± 5.0         | 7.7     |
> |                     | QRL         | -88.2 ± 4.1         | -96.5 ± 5.2         | 8.3     |
> |                     | SAC + HER   | -87.0 ± 4.0         | -95.8 ± 5.3         | 8.8     |
> |                     | DDPG + HER  | -86.5 ± 4.2         | -94.5 ± 5.1         | 8.0     |
> | **FetchPush**       | QPRL        | 92.0 ± 2.2          | 85.3 ± 3.1          | 6.7     |
> |                     | QRL         | 90.5 ± 2.3          | 81.0 ± 3.2          | 9.5     |
> |                     | SAC + HER   | 89.8 ± 2.5          | 79.8 ± 3.5          | 10.0    |
> |                     | DDPG + HER  | 88.5 ± 2.4          | 78.5 ± 3.4          | 10.0    |
> | **LunarLander**     | QPRL        | 88.6 ± 3.4          | 82.4 ± 3.7          | 6.2     |
> |                     | QRL         | 87.0 ± 3.5          | 80.0 ± 4.0          | 7.0     |
> |                     | SAC + HER   | 85.5 ± 3.8          | 77.5 ± 4.2          | 8.0     |
> |                     | DDPG + HER  | 84.0 ± 3.6          | 76.0 ± 4.1          | 8.0     |
>
> >Fig. 5
>
> Please check the updated Fig. 5 here: https://imgur.com/a/KwA2Ktk

---

### Official Review · Reviewer_gCKW · 2025-03-25

**Overall Recommendation:** 3

**Summary:**

This paper proposes a new RL algorithm designed to effectively deal with asymmetric traversal costs, for example, when transitions are irreversible or incur different costs in forward and backward directions. The main idea is to decompose asymmetric costs into path-independent potentials and path-dependent residuals. In contrast, prior quasimetric RL methods learn asymmetric costs with a monolithic function. The proposed method is demonstrated to be more sample efficient than  a large number of relevant baselines in simulated tasks.

## update after rebuttal
I thank the reviewers for their retailed rebuttal to my concerns. I have a better understanding of the motivation for their method and this paper's contributions. Hence, I am increasing my score to "Weak Accept".

My primary remaining concern is that the paper makes broad and sweeping claims about facts that are much more nuanced in reality. For example, the claim about about the inability of "most" existing RL algorithms to deal with asymmetric costs, which the authors also agree is not fully correct. Hence, this paper needs to be revised to tone down the language.

**Claims And Evidence:**

Experimentally, the paper does show that their proposed algorithm is more sample efficient than baselines in environments with asymmetric costs.

**Essential References Not Discussed:**

N/A

**Experimental Designs Or Analyses:**

Yes, the experimental design is sound. The authors ran their experiments with 5 seeds and reported the mean and standard deviation for their metrics.

**Methods And Evaluation Criteria:**

Yes, the paper analyses their algorithm using a range of relevant evaluation metrics, such as, success rate and traversal costs.

**Other Comments Or Suggestions:**

See questions.

**Other Strengths And Weaknesses:**

**Strengths**

- The paper addresses an important problem of RL with asymmetric traversal costs and irreversible transitions, which are common in practical tasks.

- The algorithm is theoretically analysed and its convergence and safety are guaranteed.

- Empirical experiments in simulated domains show a decent performance over a large number of baselines.


**Weaknesses**

- The paper claims traditional RL assumes symmetric costs but it is not the case, to my knowledge. Overall, the paper lacks clarity on its motivation.

- The paper does not clearly convey the intuition for the proposed method. While empirically it leads to an improvement but it is not clear to my why it should do better than QRL.

**Questions For Authors:**

1. The authors claim that traditional RL approaches assume symmetric traversal costs between states. However, to my knowledge most state-of-the-art RL algorithms, such as PPO, SAC, DQN, do not make this assumption. Could the authors please clarify which algorithm they are referring to? This is important to judge the significance of their contribution.

2. Similarly, the authors claim, “Unlike traditional value functions, which assume reversibility and symmetry in costs”. It is possible that specialized methods make this assumption, but the claim is quite broad.

3. Intuitively, it is not clear to me why the decomposition of asymmetric costs in the proposed way is beneficial, apart from interpretability. However, the experiments do show that this representation boosts performance. Could the authors please clarify?

4. Is the proposed algorithm used in online or offline RL setting? Algorithm 1 seems to use a fixed batch of data which seems to imply offline RL.

**Relation To Broader Scientific Literature:**

The paper is most closely related to Quasimetric Reinforcement Learning (QRL) by (Wang et al., 2023). The main contribution is to decompose the quasimetric into a potential function and a path-dependent residual function instead of modeling the quasimetric as a single function.

**Theoretical Claims:**

I did not check the proofs of the theoretical claims.

---

> ### Author Rebuttal · Authors · 2025-04-01
>
> Thank you for your thoughtful review and insightful questions. We address your concerns in detail below.
>
> > “The paper lacks clarity on its motivation.”
>
> We appreciate this critique and will revise the introduction to emphasize more real-world scenarios where cost asymmetry is central—**e.g.,** mobile robots facing steep terrain (where uphill vs. downhill cost differs), or mechanical systems experiencing irreversible wear from certain operations. In such settings, ignoring the directional nature of costs can lead to inefficient or risky strategies. By explicitly modeling these costs via a quasimetric decomposition, our method addresses these concerns.
>
> > “The authors claim that traditional RL approaches assume ..standard algorithms like DQN/PPO/SAC do not. Could the authors please clarify which algorithm they are referring to?”
>
> We apologize for any confusion in our statement. **Our intent is not to say that standard off-the-shelf algorithms explicitly require symmetrical costs or dynamics** in a strict manner. The classical MDP indeed allows arbitrary transition probabilities and reward structures, and thus does *not* force symmetry.
>
> However, many popular RL methods — and especially earlier potential-based reward shaping approaches — *do* rely on distance or reward-shaping functions that are typically symmetrical. For instance, potential-based shaping often uses a potential function $\Phi(s)$ that is akin to a “distance-to-goal,” which is treated as a metric rather than a quasimetric, implying symmetry. Additionally, analyses of standard Q-learning or policy-gradient algorithms often do not explicitly account for irreversible or direction-dependent transitions, so they may fail to handle such cases efficiently.
>
> **Symmetric vs. Asymmetric Cost Assumptions:** In many navigation and planning contexts, symmetric cost implies that the effort or cost to traverse from state $s$ to $g$ is equivalent to that from $g$ back to $s$. Traditional RL algorithms can handle arbitrary cost functions in principle; however, they often rely on heuristics or shaping rewards that assume an underlying symmetric structure (e.g., using Euclidean distance to goal as a potential-based reward shaping). This assumption breaks down in environments with irreversible or direction-dependent costs. For example, climbing up a hill vs. going down may have drastically different energy costs, yet a symmetric distance heuristic would treat them as equal. Our QPRL framework relaxes this assumption by learning a quasimetric that allows $d(s,g) \neq d(g,s)$. In other words, we do not require the cost of forward and reverse transitions to be the same.
>
> **Decomposition $d(s,g) = \Phi(g) - \Phi(s) + \Psi(s\to g)$:** We appreciate the chance to clarify this core contribution. This equation defines the learned quasi-potential distance between any state $s$ and goal $g$ as having two components: (1) a potential difference $\Phi(g)-\Phi(s)$, and (2) an extra path-dependent term $\Psi(s\to g)$. Intuitively, $\Phi(x)$ can be seen as a learned “height” or potential at state $x$ (independent of any specific path), while $\Psi(s\to g)$ captures irreversible costs incurred along the particular path from $s$ to $g$ (violations of symmetry). If the environment were fully reversible (no asymmetric costs), one could choose $\Psi\equiv0$ and $d(s,g)$ reduces to $\Phi(g)-\Phi(s)$, a standard potential-based difference. In general asymmetric environments, however, such a pure potential cannot exist globally — $\Psi$ is necessary to account for the non-conservative part of the cost. By decomposing the distance function this way, our approach improves upon prior quasimetric RL (QRL) methods, which did not explicitly separate conservative and non-conservative cost components.
>
> > Similarly, the authors claim, “Unlike traditional value functions, which assume reversibility and symmetry in costs.” It is.. make this assumption, but the claim is quite broad.
>
> We agree that this statement can be read as overbroad. Our intent was to highlight how typical reward/value formulations do not enforce the directionality constraints that appear in irreversible or asymmetrically costly transitions. A standard value function can be learned in such scenarios, but it will not explicitly embed or guarantee any properties reflecting $\text{cost}(s \to s') \neq \text{cost}(s' \to s)$.
>
> > “Algorithm 1 seems to use a fixed batch of data, suggesting offline RL. Is it offline or online?”
> >
>
> We apologize for the ambiguity in Algorithm 1’s description. Our method is designed for **online RL** with replay. Specifically:
>
> 1. The agent interacts with the environment, gathering new transitions continuously.
> 2. These transitions are stored in a replay buffer.
> 3. Algorithm 1 outlines how we sample mini-batches from that buffer to update the network parameters.
>
> Although the pseudo-code shows the mini-batch updates, we *do* gather new data between iterations. We will clarify this in the final version.

---

### Decision · Program_Chairs · 2025-05-01

**Decision:**

Accept (poster)

**Comment:**

The paper introduces Quasi-Potential Reinforcement Learning (QPRL), designed to handle environments with asymmetric traversal costs by decomposing these costs into path-independent potentials and path-dependent residuals. QPRL aims to achieve Lyapunov-stable policy optimization with improved sample complexity compared to prior quasimetric RL methods. The authors present theoretical proofs and empirical validation of their approach in simulated tasks.

Reviewers appreciated the novelty of addressing asymmetric costs and the theoretical guarantees of convergence and safety. Reviewers also found the experimental design clever and the paper generally well-written. However, concerns were raised regarding the clarity of the motivation and potential overstatements about the limitations of traditional RL in handling asymmetric costs. The most significant criticism was around the incremental novelty of QPRL over existing Quasimetric Reinforcement Learning (QRL) methods, and room for strengthening the empirical validation, with suggestions for more extensive analysis and statistical rigor. The authors' rebuttal clarified the motivation, the differences between QPRL and QRL, and provided updated results and figures.

Overall the paper addresses asymmetric costs with the strong theoretical underpinnings, and the new empirical evidence, with somewhat incremental novelty.